# *Magel2* knockdown in hypothalamic POMC neurons innervating the medial amygdala reduces susceptibility to diet-induced obesity

Yuna Choi[1,2,*], Hyeon-Young Min[1,2,*], Jiyeon Hwang[1,2], Young-Hwan Jo[1,2,3]

Hyperphagia and obesity profoundly affect the health of children with Prader–Willi syndrome (PWS). The *Magel2* gene among the genes in the Prader–Willi syndrome deletion region is expressed in proopiomelanocortin (POMC) neurons in the arcuate nucleus of the hypothalamus (ARC). Knockout of the *Magel2* gene disrupts POMC neuronal circuits and functions. Here, we report that loss of the *Magel2* gene exclusively in ARC[POMC] neurons innervating the medial amygdala (MeA) causes a reduction in body weight in both male and female mice fed with a high-fat diet. This anti-obesity effect is associated with an increased locomotor activity. There are no significant differences in glucose and insulin tolerance in mice without the *Magel2* gene in ARC[POMC] neurons innervating the MeA. Plasma estrogen levels are higher in female mutant mice than in controls. Blockade of the G protein–coupled estrogen receptor (GPER), but not estrogen receptor-α (ER-α), reduces locomotor activity in female mutant mice. Hence, our study provides evidence that knockdown of the *Magel2* gene in ARC[POMC] neurons innervating the MeA reduces susceptibility to diet-induced obesity with increased locomotor activity through activation of central GPER.

## Introduction

Prader–Willi syndrome (PWS) is a neurogenetic disorder caused by the loss of paternal expression of a cluster of genes in the human 15q11-q13 and mouse 7C regions (Boccaccio et al, 1999; Lee et al, 2000; Resnick et al, 2013). A classic sign of PWS is a constant craving for food, resulting in rapid weight gain in early childhood (Cassidy et al, 2012). Namely, insatiable appetite and hyperphagia are one of the factors that most affect children with PWS. Among the genes in the PWS deletion region, the *Magel2* gene appears to be one of the genes responsible for the etiology of PWS. Loss of the mouse

*Magel2* gene causes abnormalities in growth and fertility and increased adiposity with altered metabolism in adulthood, consistent with some of the pathologies observed in PWS (Bischof et al, 2007; Mercer & Wevrick, 2009). Although the exact cellular mechanisms underlying the abnormalities remain determined, a recent study shows deficits in secretory granule abundance and neuropeptide production in the hypothalamus of *Magel2*-null mice (Chen et al, 2020). For instance, *Magel2*-null mice have lower levels of α-melanocyte–stimulating hormone (α-MSH) derived from proopiomelanocortin (POMC) (Chen et al, 2020). As the arcuate nucleus of the hypothalamus (ARC) highly expresses *Magel2* mRNA (Kozlov et al, 2007; Maillard et al, 2016; Chen et al, 2020), a loss of function of the *Magel2* gene may disrupt the central melanocortin system, causing impairments in whole-body energy metabolisms.

POMC neurons in the ARC play a significant role in regulating food intake and body weight (Plum et al, 2006; Bumaschny et al, 2012; Greenman et al, 2013; Lam et al, 2015; Yeo et al, 2021). Acute stimulation of a subset of ARC[POMC] neurons reduces feeding (Jeong et al, 2018; Wei et al, 2018). For instance, activation of temperature-sensitive transient receptor potential cation channel subfamily V member 1 (TRPV1)–expressing POMC neurons decreases feeding (Jeong et al, 2018) and optogenetic stimulation of mature ARC[POMC] neurons in POMC[Cre] mice also lowers food intake (Wei et al, 2018). In contrast, ablation of ARC[POMC] neurons in adult mice results in hyperphagia and obesity (Zhan et al, 2013). It has been shown that *Magel2*-null mice exhibit impaired POMC neural circuits and functions (Mercer et al, 2013; Maillard et al, 2016; Oncul et al, 2018). There are fewer ARC[POMC] neurons in *Magel2*-null mice than in controls (Mercer et al, 2013). ARC[POMC] axonal projections to the paraventricular (PVN) and dorsomedial (DMH) hypothalamus are significantly reduced by *Magel2* deletion (Maillard et al, 2016). Furthermore, the basal spontaneous activity of ARC[POMC] neurons is lower in *Magel2*-null mice than in controls (Oncul et al, 2018). Hence, disrupted ARC[POMC] neural circuits may result in changes in metabolic homeostasis in *Magel2*-null mice.

[1]Fleischer Institute for Diabetes and Metabolism, Albert Einstein College of Medicine, New York City, NY, USA   [2]Division of Endocrinology, Department of Medicine, Albert Einstein College of Medicine, New York City, NY, USA   [3]Department of Molecular Pharmacology, Albert Einstein College of Medicine, New York City, NY, USA

Correspondence: young-hwan.jo@einsteinmed.edu
*Yuna Choi and Hyeon-Young Min contributed equally to this work.

Among the downstream targets of ARC[POMC] axonal projections, POMC projections to the PVN and medial amygdala (MeA) are implicated in controlling feeding behavior through activation of the melanocortin receptor type 4 (MC4R) (Balthasar et al, 2005; Liu et al, 2013; Shah et al, 2014; Kwon & Jo, 2020). We recently demonstrated that optogenetic stimulation of the ARC[POMC]→MeA pathway decreased liquid food intake blocked by the MC4R antagonist (Kwon & Jo, 2020). Given that the ARC[POMC]→MeA pathway regulates feeding, we specifically examined the role of the *Magel2* gene in ARC[POMC] neurons innervating the MeA in controlling food intake and body weight gain. Unexpectedly, knockdown of the *Magel2* gene in MeA-projecting ARC[POMC] neurons reduced rather than increased body weight without changing food intake in mice when fed a high-fat diet (HFD). Importantly, this anti-obesity effect was associated with increased locomotor activity in male and female mice. Our results provide cellular evidence that MAGEL2 in MeA-projecting ARC[POMC] neurons plays an essential role in controlling energy balance.

# Results

### ARC[POMC] neurons express MAGEL2

High expression of *Magel2* mRNAs was detected in the ARC (Kozlov et al, 2007; Maillard et al, 2016; Chen et al, 2020), but the neurochemical identity of *Magel2*-expressing neurons in the area remains unknown. We first examined if ARC[POMC] neurons express MAGEL2. We used POMC[Cre:Rosa26-GFP] mice, in which the POMC[CRE] transgene causes cell-specific recombination to induce expression

of eGFP from the Rosa26 promoter and an antibody directed against MAGEL2 (Chen et al, 2020). MAGEL2 expression was detected in the ARC as described previously (Chen et al, 2020). We found that more than 60% of ARC[POMC] neurons in male POMC[Cre:Rosa26-GFP] mice were positive for MAGEL2 (n = 1,771 of 2,756 POMC neurons; Fig 1A and B). Similarly, most of the ARC[POMC] neurons were labeled with the anti-MAGEL2 antibody in female POMC[Cre:Rosa26-GFP] mice (n = 3,265 of 4,543 neurons; Fig 1C and D). Our immunostaining results support the interpretation that MAGEL2 may play a role in controlling ARC[POMC] neuron functions.

### Knockdown of the *Magel2* gene exclusively in MeA-innervating ARC[POMC] neurons causes a reduction in body weight in male mice fed with a high-fat diet

PWS animal models, such as *Magel2*- and *Snord116*-null mice fed with a standard chow diet did not develop the delayed-onset obesity described in PWS (Bischof et al, 2007; Qi et al, 2016). Interestingly, although the overall body weight in *Magel2*-null mice was not different from that in the control group, *Magel2*-null mice exhibited a significant increase in adiposity (Bischof et al, 2007). We thus sought to determine if the nutrient excess induces diet-induced obesity in our animal model. To investigate the role of the *Magel2* gene in ARC[POMC] neurons innervating the MeA in controlling energy balance, we knocked down the *Magel2* gene exclusively in ARC[POMC] neurons that project to the MeA with the use of CRISPR-Cas9 technology as described in our prior study (Jeong et al, 2018). We crossbred the POMC[Cre] strain with the floxed-stop Cas9-eGFP strain to generate POMC[Cre;Cas9-eGFP] mice. We bilaterally

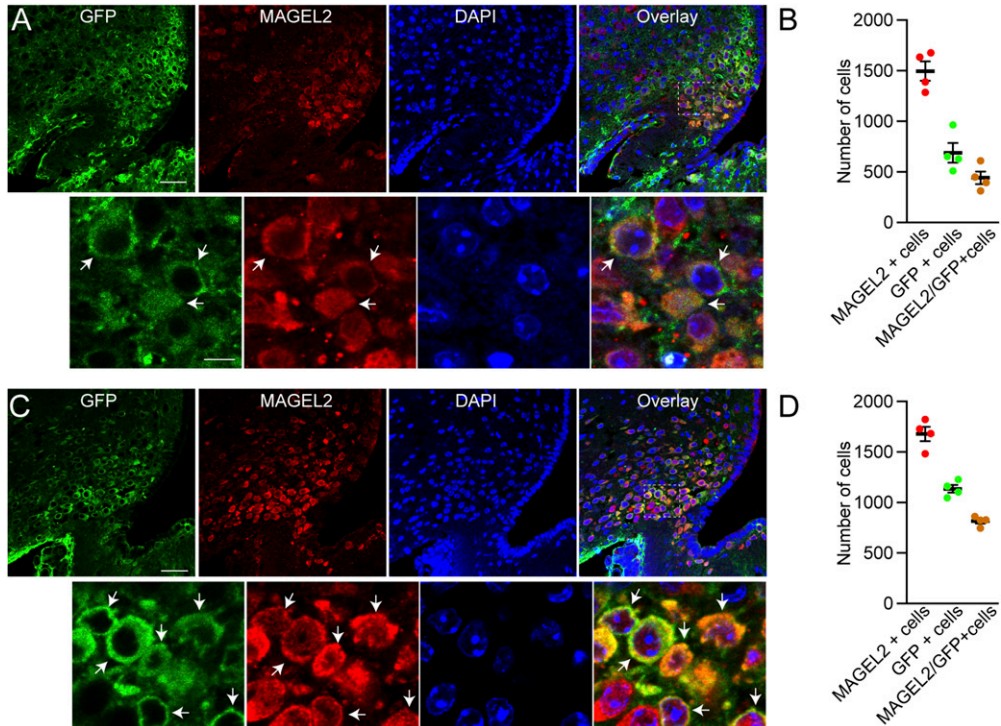

**Figure 1. ARC[POMC] neurons express MAGEL2.**

**(A)** Images of confocal fluorescence microscopy showing double immunostaining with anti-GFP (green) and anti-MAGEL2 (red) antibodies in the ARC of male POMC[Cre:GFP] mice. Scale bar: 30 μm. Bottom panel: higher magnification view of neurons co-expressing GFP and MAGEL2 (white arrows). Scale bar: 10 μm. **(B)** Summary plot showing the number of MAGEL2-positive POMC neurons in the ARC of POMC[Cre:GFP] mice (n = 4 mice). **(C)** Images of confocal fluorescence microscopy showing double immunostaining with anti-GFP (green) and anti-MAGEL2 (red) antibodies in the ARC of female POMC[Cre:GFP] mice. Scale bar: 30 μm. Bottom panel: higher magnification view of neurons co-expressing GFP and MAGEL2 (white arrows). Scale bar: 10 μm. **(D)** Summary plot showing the number of MAGEL2-positive POMC neurons in the ARC of POMC[Cre:GFP] mice (n = 4 mice).

injected a retrograde adeno-associated virus (AAV) encoding mouse *Magel2* single guide RNA (retroAAV-*Magel2* sgRNA) into the MeA of POMC[Cre:Cas9-eGFP] and POMC[Cre] mice (Figs 2A and S1). After viral injection, mice were given a high-fat diet for 10 wk. To validate our experimental approach, the *Magel2* gene expression in the ARC at 10 wk post-viral injections was analyzed by reverse transcription-quantitative polymerase chain reaction (RT-qPCR). We found that male POMC[Cre:Cas9-GFP] mice receiving retroAAV-*Magel2* sgRNA exhibited lower *Magel2* expression in the ARC compared with control mice (Fig 2A). In addition to reduced *Magel2* gene expression, the number of MAGEL2-positive cells in the ARC decreased in POMC[Cre:Cas9-GFP] mice receiving retroAAV-*Magel2* sgRNA injection (Fig 2B), demonstrating the feasibility and efficacity of retroAAV-*Magel2* sgRNA.

When fed HFD for 10 wk, we found that male POMC[Cre:Cas9-GFP] mice receiving retroAAV-*Magel2* sgRNA significantly gained less body weight than controls (Fig 2C). Lower body weight was associated with a reduction in body fat mass but not lean mass (Fig 2D and E). We next asked if this body weight loss is attributed to either reduced energy intake, increased energy expenditure, or both. Although body weight was significantly different during high-fat feeding, daytime and nighttime food consumption between the experimental and control groups was not significantly different (Fig 2F).

The absence of reduced food intake suggests that changes in energy expenditure may result in weight loss. We investigated if there is enhanced energy expenditure resulting from either enhanced basal metabolic rate and/or physical activity. We placed mice in metabolic cages to measure energy expenditure (measured as $O_2$ consumption [$VO_2$] and respiratory exchange ratio [RER]). There was no significant difference in $O_2$ consumption and RER between the two groups (Fig 2G and H). Interestingly, nighttime locomotor activity in male POMC[Cre:Cas9-GFP] mice receiving retroAAV-*Magel2* sgRNA was higher than that in the control group (Fig 2I and J). Unexpectedly, total energy expenditure between the groups was similar (Fig 2K–M). Hence, our results revealed that the loss of function of the *Magel2* gene in ARC[POMC] neurons innervating the MeA resulted in increased physical activity.

Given that POMC[Cre:Cas9-eGFP] mice receiving retroAAV-*Magel2* sgRNA injection exhibited lowered body weight gain, we sought to determine if knockdown of the *Magel2* gene improves glucose homeostasis. We first quantified basal (non-fasting) and fasting glucose levels. There were no significant differences in basal and fasting blood glucose levels between the groups (Fig 3A and B). And then, we performed glucose tolerance tests to assess the ability of male POMC[Cre:Cas9-GFP] mice receiving retroAAV-*Magel2* sgRNA injection to dispose of a glucose load. *Magel2* knockdown did not change glucose clearance (Fig 3C). We also carried out insulin tolerance tests to assess glucose levels over time to an i.p. insulin injection and found that insulin tolerance was not affected by *Magel2* knockdown (Fig 3D). Hence, lower body weight in male POMC[Cre:Cas9-GFP] mice receiving retroAAV-*Magel2* sgRNA injection did not occur in conjunction with altered glucose homeostasis. We analyzed plasma levels of leptin and insulin that play a key role in controlling satiety, insulin sensitivity, and glucose homeostasis. *Magel2* knockdown had no effects on plasma leptin and insulin levels (Fig 3E and F).

## Knockdown of the *Magel2* gene exclusively in MeA-innervating ARC[POMC] neurons causes a reduction in body weight in female mice fed with a high-fat diet

As *Magel2* knockdown in ARC[POMC] neurons innervating the MeA lowered body weight gain in male mice, we examined if female mice without the *Magel2* gene in ARC[POMC] neurons innervating the MeA similarly do not develop DIO. Injection of retroAAV-*Magel2* sgRNA into the MeA of female POMC[Cre:Cas9-GFP] mice significantly decreased expression of the *Magel2* gene and protein in the ARC (Fig 4A and B). Likewise, female mice receiving retroAAV-*Magel2* sgRNA injection gained less body weight than controls during high-fat feeding (Fig 4C). Fat mass in mutant female mice significantly differed from that in controls, whereas lean body mass was similar between the experimental groups (Fig 4D and E).

We also found that knockdown of the *Magel2* gene in female mice did not change food consumption (Fig 4F). Both groups consumed a similar amount of HFD during the day and dark phases. In addition, measurement of $O_2$ consumption revealed that female POMC[Cre:Cas9-GFP] mice receiving retroAAV-*Magel2* sgRNA also exhibited no significant difference in $VO_2$ compared with the control group, although there was a trend toward an increase in $VO_2$ in the mutant mice (Fig 4G). The RER was not significantly different between the experimental groups (Fig 4H). Interestingly, similar to males, total locomotor activity in the dark phase was significantly higher in female POMC[Cre:Cas9-GFP] mice receiving retroAAV-*Magel2* sgRNA compared with that in the control group (Fig 4I and J). In addition, female mice without the *Magel2* gene in ARC[POMC] neurons innervating the MeA exhibited higher daytime locomotor activity compared with the controls (Fig 4I and J). A trend toward an increase in TEE was observed in female mice receiving retroAAV-*Magel2* sgRNA injection (Fig 4K–M). Hence, the loss of function of the *Magel2* gene in ARC[POMC] neurons caused increased physical activity in female mice as well.

We also sought to determine if knockdown of the *Magel2* gene improves glucose homeostasis. The experimental and control groups exhibited no significant differences in basal and fasting blood glucose levels (Fig 5A and B). We found no alterations in glucose and insulin tolerance (Fig 5C and D). No significant differences were found in plasma insulin levels between the two groups, but the mutant mice showed a trend of lower levels of leptin (Fig 5E and F). Collectively, our results suggest that increased locomotor activity contribute to body weight loss after *Magel2* knockdown in ARC[POMC] neurons innervating the MeA in female mice as well.

## Knockdown of the *Magel2* gene in MeA-projecting ARC[POMC] neurons elevates plasma estrogen levels in female mice

Estrogen played a critical role in driving physical activity in female mice through activation of the nuclear estrogen receptor-α (ER-α) in the brain, including the hypothalamus and MeA (Ogawa et al, 2003; Xu et al, 2011, 2015; Krause et al, 2021). In addition, ER-α KO mice exhibited increased adiposity, insulin resistance, and impaired glucose tolerance in males and females (Heine et al, 2000). Central ER-α KO mice also displayed hyperphagia, increased body weight, adiposity, and reduced physical activity (Xu et al, 2011). Thus,

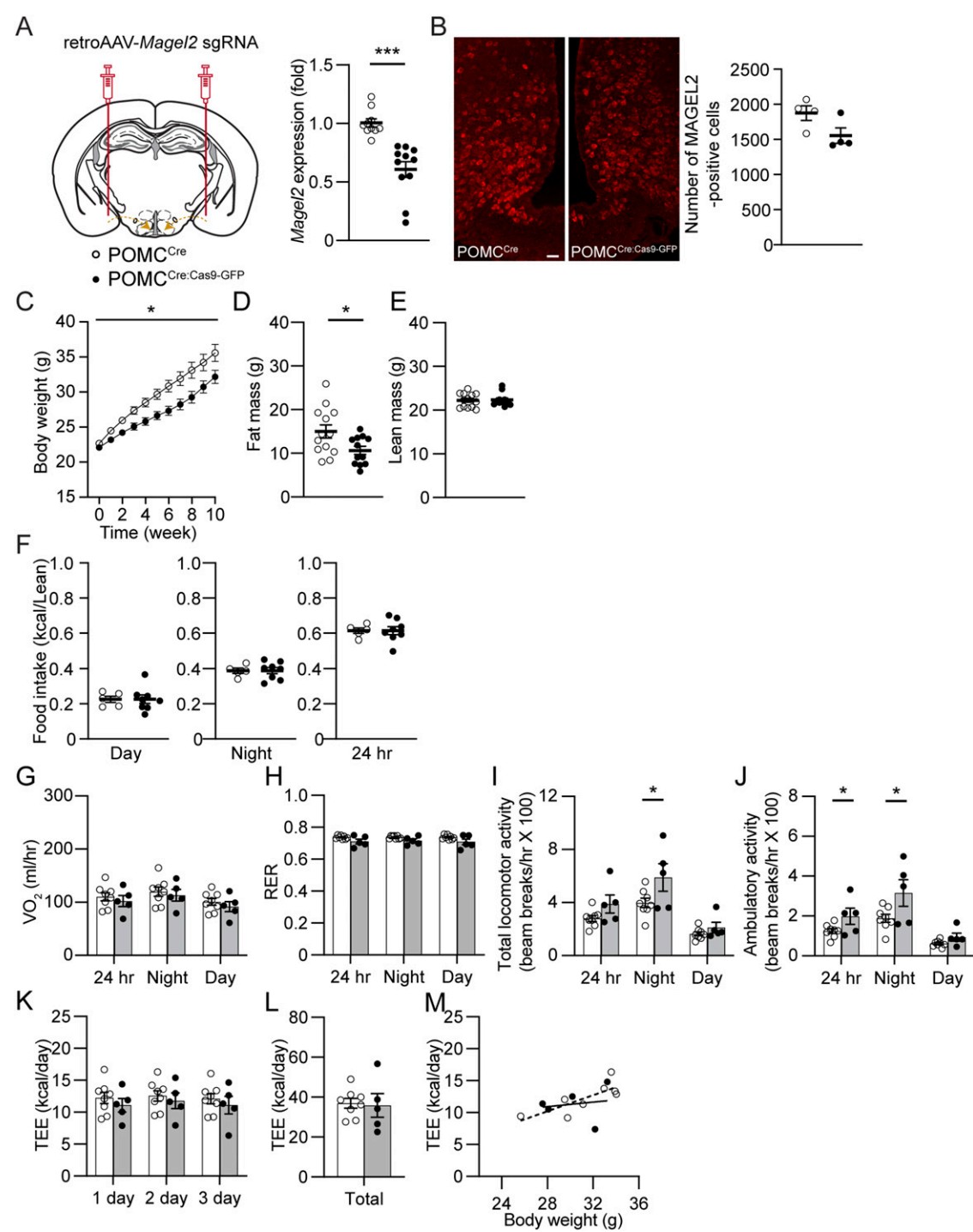

**Figure 2. Loss of the *Magel2* gene in ARC[POMC] neurons innervating the MeA causes a reduction in body weight while increasing locomotor activity in male mice fed with HFD.**
**(A)** Schematic diagram of the experimental configuration. Retrograde AAV-*Magel2* sgRNA viruses were bilaterally injected into the MeA of POMC[Cre] and POMC[Cre:Cas9-GFP] mice. Left panel: Summary plot showing relative expression of the *Magel2* gene in the ARC of POMC[Cre] (open circle; n = 10 mice) and POMC[Cre:Cas9-GFP] (closed circle; n = 11 mice) mice receiving retrograde AAV-*Magel2* sgRNA viral injection to the MeA. There was a significant difference in *Magel2* expression between the two groups. Two-tailed *t* test, ***$P < 0.001$. **(B)** Images of confocal fluorescence microscopy showing MAGEL2 expression in the ARC of POMC[Cre] (left) and POMC[Cre:Cas9-GFP] (right) mice receiving retrograde AAV-*Magel2* sgRNA viral injection to the MeA. Scale bar, 30 $\mu$m. Right panel: Summary plot showing the number of MAGEL2-positive cells in the ARC of POMC[Cre] (left; n = 4 mice) and POMC[Cre:Cas9-GFP] (right; n = 4 mice) mice receiving retrograde AAV-*Magel2* sgRNA viral injection. **(C)** Pooled data of body weight obtained from POMC[Cre] (open circle; n = 17 mice) and POMC[Cre:Cas9-GFP] (closed circle; n = 14 mice) mice receiving retrograde AAV-*Magel2* sgRNA viral injection to the MeA. The loss of the *Magel2* gene in ARC[POMC] neurons innervating the MeA resulted in body weight loss in male mice fed with HFD for 10 wk. Two-way repeated measures ANOVA followed by Sidak multiple comparisons test (between the groups, $F_{(1, 16)} = 8.1$, *$P < 0.05$). **(D, E)** Pooled data of body composition from POMC[Cre] (n = 13 mice) and POMC[Cre:Cas9-GFP] (n = 12 mice)

we hypothesized that higher locomotor activity and altered energy expenditure in POMC[Cre:Cas9-GFP] mice receiving retroAAV-*Magel2* sgRNA may be due to enhanced estrogen/ER-α signaling in the brain. We thus examined if *Magel2* knockdown in ARC[POMC] neurons innervating the MeA elevates plasma estrogen levels. We found that plasma estrogen levels were significantly higher in female POMC[Cre:Cas9-GFP] mice receiving retroAAV-*Magel2* sgRNA injection than in the control group (Fig 6A). Although we failed to detect plasma estrogen in male mice because their estrogen levels were outside the detection range, levels of plasma testosterone in male POMC[Cre:Cas9-GFP] mice receiving retroAAV-*Magel2* sgRNA mice did not differ from those in the control group (POMC[Cre]+*Magel2* sgRNA, 294 ± 36 pg/ml, n = 10 mice; POMC[Cre+Cas9]+*Magel2* sgRNA, 312 ± 36 pg/ml, n = 11 mice, $P > 0.05$).

Hence, we sought to determine if blockade of central ER-α could reverse the effects of *Magel2* knockdown on locomotor activity. We infused the ER-α antagonist ICI 182,780 (Fulvestrant; 40 ng/day) into the lateral ventricle via osmotic pumps and placed mice in metabolic cages. We chose the dose based on previous studies, in which that dose of the ER-α antagonist completely blocked the effect of central ER-α (Xue et al, 2007, 2008; Stell et al, 2008). Infusion of the ER-α antagonist did not change $VO_2$ and RER (Fig 6B and C). In contrast to our expectation, female POMC[Cre:Cas9-GFP] mice receiving retroAAV-*Magel2* sgRNA injection responded to ICI 182,780 with a significant increase rather than decrease in physical activity in the dark phase (Fig 6D and E). There was no significant difference in total energy expenditure between the groups (Fig 6F–H). These unexpected findings suggest that activation of the nuclear ER-α may not increase physical activity in female POMC[Cre:Cas9-GFP] mice receiving retroAAV-*Magel2* sgRNA injection.

### Central estrogen-GPER interaction causes increased locomotor activity in female mice

In addition to the nuclear ER-α which predominantly regulates transcription (Fuentes & Silveyra, 2019), the G-protein-coupled estrogen receptor (GPER; previously known as GPR30) was found in the brain and played a role in metabolic regulation (Prossnitz & Barton, 2011). Although the ER-α antagonist ICI 182,780 has been widely used to treat breast cancer in postmenopausal women (Nathan & Schmid, 2017), it is described that this substance can also activate GPER (Filardo et al, 2000; Meyer et al, 2010). It is thus possible that activation of central GREP may promote physical activity in our preparations. We first carried out immunostaining with an anti-GPER antibody to investigate if GPER is expressed in the VMH and MeA as these structures are implicated in estrogen-induced physical activity in mice (Xu et al, 2015; Krause et al, 2021). Immunostaining revealed that GPER was detected in the VMH as well as the MeA (Fig 7A), suggesting that increased estrogen levels may elevate locomotor activity via activation of central GPER.

Next, we sought to determine if blockade of central GPER could reduce increased locomotor activity observed in female mice receiving retroAAV-*Magel2* sgRNA injection to the MeA. The potent GPER antagonist G15 was infused into the lateral ventricle via osmotic pumps. Like ICI 182,780, the GPER antagonist did not alter $VO_2$ and RER (Fig 7B and C). However, in contrast to the effects of ICI 182,780 on physical activity, we found that female POMC[Cre:Cas9-GFP] mice receiving retroAAV-*Magel2* sgRNA injection did not exhibit increased locomotor activity (Fig 7D and E). There was no significant difference in TEE (Fig 7F–H). Hence, our results suggest that GPER expressed in the VMN and MeA may contribute to the control of locomotor activity in female POMC[Cre:Cas9-GFP] mice receiving retroAAV-*Magel2* sgRNA injection.

## Discussion

Our current study provided physiological evidence for the role of the *Magel2* gene in ARC[POMC] neurons innervating the MeA in the control of energy balance. Prior studies demonstrated *Magel2* mRNA expression in the hypothalamus, particularly the ARC (Kozlov et al, 2007; Maillard et al, 2016; Chen et al, 2020). We found that most of the ARC[POMC] neurons expressed MAGEL2. The knockdown of the *Magel2* gene exclusively in ARC[POMC] neurons innervating the MeA protected against DIO in both male and female mice, consistent with prior studies with PWS mouse models, including mice lacking the *Magel2* and *Snord116* genes (Bischof et al, 2007; Qi et al, 2016). The cellular mechanisms underlying the anti-obesity effects appear to be sex-dependent. Both male and female mutant mice exhibited higher locomotor activity than the control groups. Interestingly, it seems likely that sex hormone levels played a key role in regulating locomotor activity in females. In fact, there was a significant increase in estrogen levels in female POMC[Cre:Cas9-GFP] mice receiving retroAAV-*Magel2* sgRNA injection. In contrast, no difference in testosterone levels was found between the male groups. Finally, blockade of GPER but not ER-α completely abolished the effect of loss of the *Magel2* gene in ARC[POMC] neurons innervating the MeA. This was observed in female mice only. Hence, central estrogen and GPER interaction played a critical role in controlling energy balance in our PWS mouse model.

Loss of imprinted genes at the PWS-chromosome domain, such as the *Magel2*, *Snord116*, and *Ndn* genes in mice, caused behavioral and neuroendocrine alterations reminiscent of PWS (Muscatelli et al, 2000; Bischof et al, 2007; Kozlov et al, 2007; Qi et al, 2016; Burnett et al, 2017). For instance, mice lacking the *Ndn* gene exhibited early post-natal lethality partly because of a respiratory defect (Gerard et al, 1999; Muscatelli et al, 2000). This finding correlated with neonatal respiratory distress observed in PWS patients (Alfaro et al, 2019). In addition, these mutant mice had fewer oxytocin neurons in the paraventricular nucleus of the hypothalamus than controls

mice receiving retrograde AAV-*Magel2* sgRNA viral injection to the MeA. Two-tailed *t* test, *$P < 0.05$. **(F)** Pooled data showing no significant difference in food intake between the groups (POMC[Cre], n = 5 mice; POMC[Cre:Cas9-GFP], n = 8 mice). **(G, H)** Summary plot showing $VO_2$ (G) and respiratory exchange ratio (H) between the groups. (POMC[Cre], n = 8 mice, POMC[Cre:Cas9-GFP], n = 5 mice). **(I, J)** Pooled data showing increased total and ambulatory activity in mice without the *Magel2* gene in ARC[POMC] neurons innervating the MeA (POMC[Cre], n = 8 mice, POMC[Cre:Cas9-GFP], n = 5 mice). Two-tailed *t* test, *$P < 0.05$. **(K, L, M)** Summary plot showing TEE between the experimental groups. Regression plot showing TEE versus body weight (M).

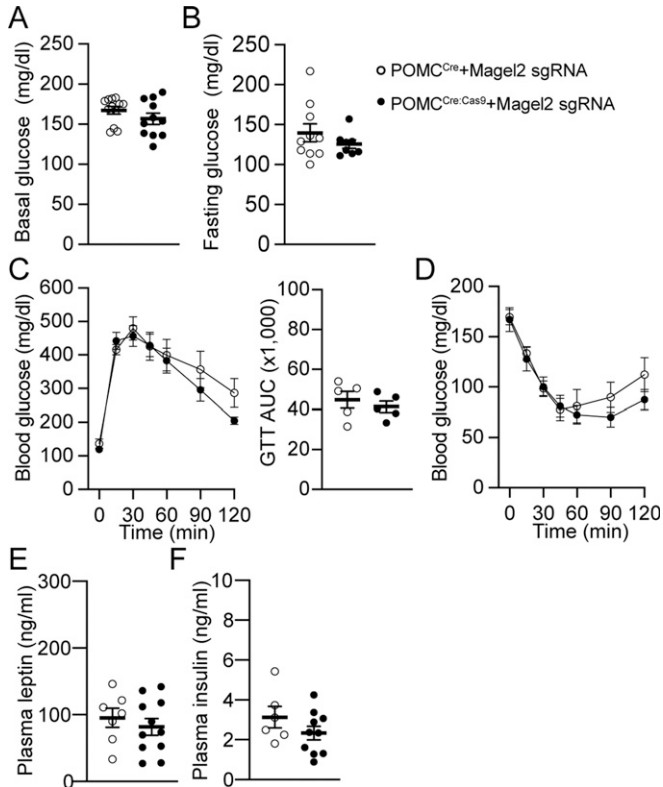

**Figure 3. Loss of the *Magel2* gene in ARC[POMC] neurons innervating the MeA does not alter glucose homeostasis in male mice fed with HFD.**
**(A, B)** Summary plots showing non-fasting (n = 12 mice versus 11 mice) and fasting (n = 10 mice versus 8 mice) blood glucose levels in POMC[Cre] (open circle) and POMC[Cre:Cas9-GFP] (closed circle) mice receiving retrograde AAV-*Magel2* sgRNA viral injection to the MeA. **(C)** Pooled data showing GTT in mice with and without the *Magel2* gene in ARC[POMC] neurons innervating the MeA (left). Right panel: graphs showing areas under the curves (AUC) values obtained from GTT experiments (two-tailed t test, POMC[Cre], n = 5 mice, POMC[Cre:Cas9-GFP], n = 5 mice). **(D)** Pooled data showing ITT in mice with and without the *Magel2* gene in ARC[POMC] neurons innervating the MeA. No significant difference was observed between the groups (POMC[Cre], n = 7 mice, POMC[Cre:Cas9-GFP], n = 5 mice). Two-way ANOVA followed by Sidak multiple comparisons test, $F_{(1,10)}$ = 0.37. **(E, F)** Summary plots showing levels of plasma leptin and insulin in mice with and without the *Magel2* gene in ARC[POMC] neurons innervating the MeA (POMC[Cre], n = 7 mice, POMC[Cre:Cas9-GFP], n = 11 mice for leptin; POMC[Cre], n = 6 mice, POMC[Cre:Cas9-GFP], n = 10 mice for insulin).

(Muscatelli et al, 2000), consistent with the findings in PWS patients (Swaab, 1997). Although it is expected that the loss of function of the PWS imprinted genes causes the development of obesity, as observed in individuals with PWS, several studies failed to show excessive body weight gain in the PWS animal models. For instance, the germline and paternal deletion of the *Snord116* gene in mice fed with a standard chow diet caused growth retardation, such as decreased body length and weight at weaning and throughout adulthood (Ding et al, 2008; Qi et al, 2016; Burnett et al, 2017). Despite their weight loss, the *Snord116* KO mice exhibited increased food intake relative to body weight (Ding et al, 2008; Qi et al, 2016; Burnett et al, 2017). In addition, mice deficient in the paternal allele of *Magel2* displayed no significant difference in body weight compared with the control group when given a normal chow (Bischof et al, 2007). Of particular interest is that both the *Snord116*-null mice

and mice deficient in the paternal allele of *Magel2* did not develop DIO when maintained on HFD (Bischof et al, 2007; Qi et al, 2016). Hence, the effects of the loss of function of individual genes in the PWS domain may differ from those of loss of the entire imprinted genes in the PWS domain. In other words, each gene in the PWS imprinted region may have a distinct function and differently contribute to the phenotypes of PWS.

The cellular mechanisms by which paternally imprinted genes in the PWS domain influence biological processes remain to be determined. Prior studies described that impaired prohormone and propeptide posttranslational modification might disrupt hypothalamic neuroendocrine function in PWS animal models (Burnett et al, 2017; Chen et al, 2020). For instance, paternal loss of the *Snord116* gene in mice down-regulated *Pcsk1* transcript and prohormone convertase 1 (PC1), the protein product of the *Pcsk1* gene in the hypothalamus, pancreatic islet, and stomach (Burnett et al, 2017). A down-regulation of PC1 reduced circulating levels of insulin, ghrelin, and growth hormone-releasing hormone (Burnett et al, 2017). Consequently, deficiencies in prohormone processing may result in the neuroendocrine phenotype of PWS, including hypogonadism, short stature, and type 2 diabetes (Burnett et al, 2017). Similarly, loss of the *Magel2* gene in mice caused decreased neuropeptide production in the hypothalamus because of down-regulation of neuropeptide processing enzymes, including PC1, PC2, and carboxypeptidase E (Chen et al, 2020). Consequently, hypothalamic and plasma levels of neuropeptides such as vasopressin, oxytocin, somatostatin, and agouti-related peptide were lower in *Magel2*-null mice than in controls (Chen et al, 2020). Interestingly, levels of α-MSH produced from the POMC precursor by POMC posttranslational modification enzymes, including PC1, PC2, and CPE were also significantly lower in *Magel2* KO mice than in controls (Chen et al, 2020). As hypothalamic α-MSH and its cognate receptors play a major role in regulating energy balance and glucose homeostasis (Wallingford et al, 2009; Shah et al, 2014; Schneeberger et al, 2015; Tooke et al, 2019), a disruption in the central melanocortin system in PWS may cause behaviral and neuroendocrine changes.

Our present study revealed that most of the ARC[POMC] neurons expressed MAGEL2, supporting the previous findings that the loss of the *Magel2* gene disrupted hypothalamic POMC neural circuits and functions (Mercer et al, 2013; Maillard et al, 2016; Oncul et al, 2018). For instance, *Magel2*-null mice exhibited altered ARC[POMC] neuron electric activity (Oncul et al, 2018), impaired ARC[POMC] neuronal projections to the PVH, ARC, and DMH (Maillard et al, 2016), and lack of leptin's anorexic effect (Mercer et al, 2013). Prior studies (Elias et al, 1998, 1999; King & Hentges, 2011; Dicken et al, 2012; Henry et al, 2015; Koch et al, 2015; Wang et al, 2015; Campbell et al, 2017; Chen et al, 2017; Lam et al, 2017; Biglari et al, 2021), including our own work (Lee et al, 2015; Jeong et al, 2016; Kwon & Jo, 2020) demonstrated that ARC[POMC] neurons were neurochemically and neuroanatomically heterogeneous. Our recent studies further showed functional heterogeneity of ARC[POMC] neurons (Jeong et al, 2018; Kwon et al, 2020; Kwon & Jo, 2020). In other words, neurochemically and neuroanatomically distinct subpopulations of POMC neurons had distinct target sites and metabolic functions.

In this study, we knocked down the expression of the *Magel2* gene exclusively in the ARC[POMC]→MeA projection because ARC[POMC] projections to the MeA controlled acute food intake through

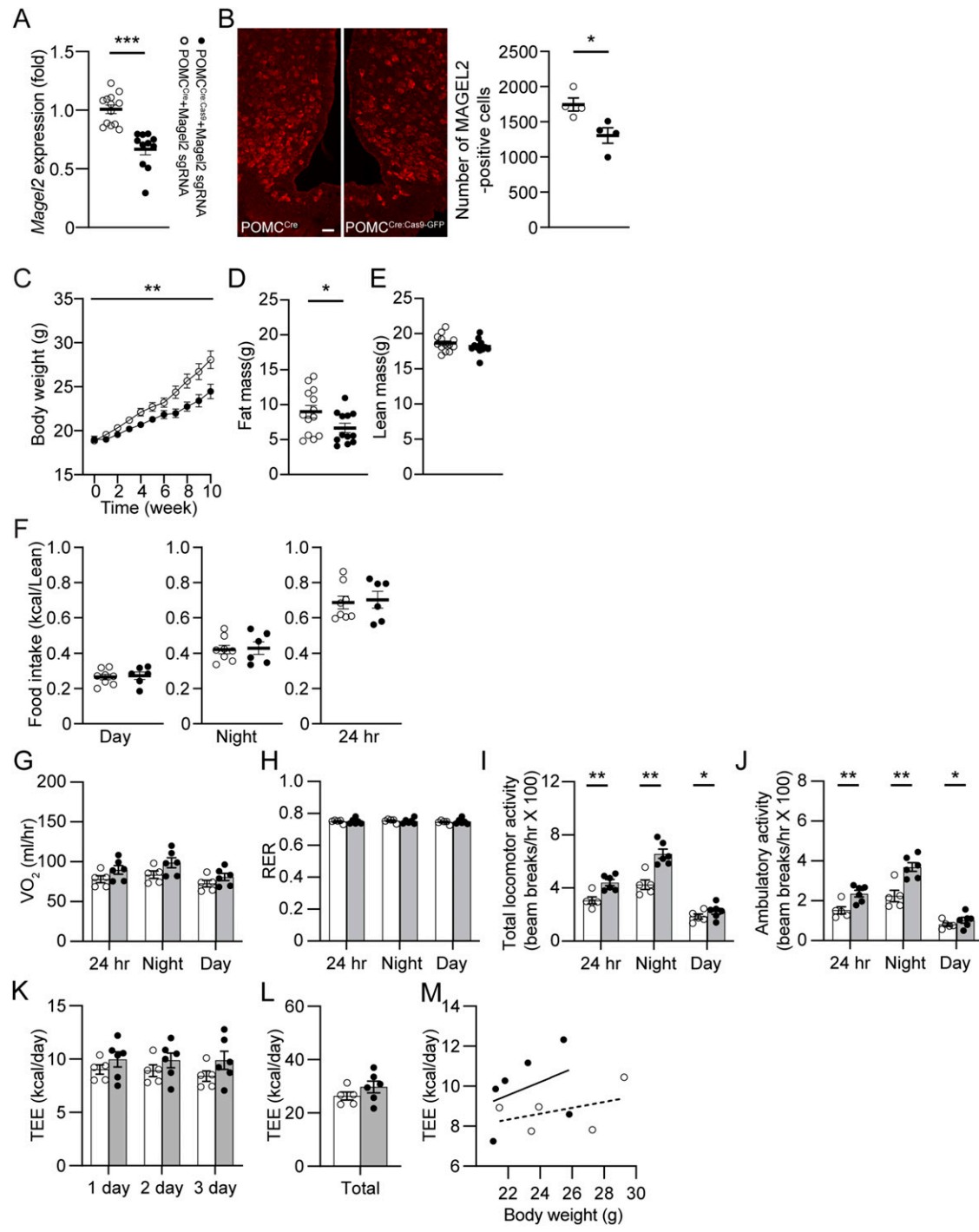

**Figure 4. Loss of the *Magel2* gene in ARC^POMC neurons innervating the MeA causes a reduction in body weight while increasing locomotor activity in female mice fed with HFD.**

**(A)** Summary plot showing relative expression of the *Magel2* gene in the ARC of POMC^Cre (open circle; n = 13 mice) and POMC^Cre:Cas9-GFP (closed circle; n = 11 mice) mice receiving retrograde AAV-*Magel2* sgRNA viral injection to the MeA. Two-tailed *t* test, ***$P < 0.001$. **(B)** Images of confocal fluorescence microscopy showing MAGEL2 expression in the ARC of mice with (left) and without (right) the *Magel2* gene in ARC^POMC neurons innervating the MeA. Scale bar, 30 μm. Right panel: Summary plot showing the number of MAGEL2–positive cells in the ARC. *$P < 0.05$. **(C)** Pooled data of body weight obtained from POMC^Cre (open circle, n = 24 mice) and POMC^Cre:Cas9-GFP (closed circle, n = 18 mice) mice receiving retrograde AAV-*Magel2* sgRNA viral injection to the MeA. A significant difference in body weight was observed between the groups. Two-way repeated measures ANOVA followed by Sidak multiple comparisons test (between the groups, $F_{(1, 40)} = 8.39$, **$P < 0.01$). **(D, E)** Pooled data of body composition from POMC^Cre (n = 13 mice) and POMC^Cre:Cas9-GFP (n = 12 mice) mice receiving retrograde AAV-*Magel2* sgRNA viral injection to the MeA. Two-tailed *t* test, *$P < 0.05$. **(F)** Pooled data showing no significant difference in food intake between the groups (POMC^Cre, n = 8 mice; POMC^Cre:Cas9-GFP, n = 6 mice). **(G, H)** Summary plot showing VO₂ (G) and respiratory exchange ratio (H) between the groups. No significant difference in VO₂ and respiratory exchange ratio was observed in mice without the *Magel2* gene in ARC^POMC neurons innervating the MeA (POMC^Cre, n = 5 mice; POMC^Cre:Cas9-GFP, n = 6 mice). **(I)** Pooled data showing increased total and ambulatory activity in mice without

activation of MC4R (Balthasar et al, 2005; Liu et al, 2013; Kwon & Jo, 2020). We expected that a loss of the *Magel2* gene in this subset of ARC[POMC] neurons would disrupt this melanocortin pathway. As a result, the HFD challenge would result in DIO in POMC[Cre:Cas9-GFP] mice receiving retroAAV-*Magel2* sgRNA injection to the MeA. Contrary to our expectation, knockdown of the *Magel2* gene in MeA-innervating ARC[POMC] neurons protected the mutant mice from DIO, consistent with the prior studies with *Snord116* KO and *Magel2* KO male animals (Bischof et al, 2007; Qi et al, 2016). We found no difference in food intake relative to lean mass between the male mutant and control groups, meaning that they did not develop hyperphagia. However, reduced body weight was associated with a decreased percentage of fat mass, consistent with the finding in *Snord116* KO mice (Qi et al, 2016). Given the reported similarity of our male mutant mice to *Snord116*-deficient mice (e.g., reduced body, no effect on food intake, decreased percentage of fat mass, increased locomotor activity at the dark phase) (Ding et al, 2008; Qi et al, 2016), it appears that both genes do not contribute to the control of energy intake. As both mutant mice exhibited increased locomotor activity, this may reduce susceptibility to DIO.

The difference in body weight between the two groups during high-fat feeding would be due partly to altered energy expenditure. Indirect calorimetry revealed no significant differences in daytime and nighttime $O_2$ consumption. Although we failed to detect a significant difference in energy expenditure between the two groups, it does not mean that there was no difference in energy expenditure between the groups during high-fat feeding. It is highly possible that our system may not be able to detect a subtle change in energy expenditure. It is also possible that an accumulation of small improvements in basal energy expenditure because of increased physical activity from the beginning of high-fat feeding may prevent weight gain in our animal model. In fact, female mice lacking the *Magel2* in ARC[POMC] neurons innervating the MeA exhibited a trend toward an increase in total energy expenditure. In addition, POMC[Cre:Cas9-GFP] male mice receiving retroAAV-*Magel2* sgRNA injection to the MeA exhibited higher locomotor activity in the dark phase compared with the control group. Locomotor activity in *Snord116*-null male mice significantly increased only in the dark phase as well (Qi et al, 2016). Thus, it is plausible that increased physical activity in male mice without imprinted genes at the PWS-chromosome domain would cause body weight loss. Although male POMC[Cre:Cas9-GFP] mice receiving retroAAV-*Magel2* sgRNA injection to the MeA were resistant to DIO, there were no improvements in glucose homeostasis and insulin sensitivity in our preparations. Basal and fasting glucose levels in the mutant mice did not differ from those in their controls, and no difference in glucose tolerance was detected between the groups. In addition, basal insulin levels and insulin tolerance were not altered by loss of the *Magel2* gene in MeA-innervating ARC[POMC] neurons.

Resistance to DIO appeared partly because of increased locomotor activity in the dark phase. What causes an increase in physical activity in PWS animal models? Prior studies demonstrated that central ER-α could control physical activity in female mice

(Ogawa et al, 2003; Xu et al, 2011, 2015; Krause et al, 2021). For example, mice lacking central ER-α exhibited reduced physical activity, resulting in increased body weight (Xu et al, 2011). Activation of ER-α in the VMH promoted physical activity in female mice via increased expression of *Mc4r* transcripts in the VMH (Krause et al, 2021). Moreover, chemogenetic stimulation of VMH[MC4R] neurons increased spontaneous physical activity in both male and female mice which was sufficient to reduce body weight (Krause et al, 2021). Selective deletion of ER-α in single-minded (SIM1) neurons in the MeA also showed a significant reduction in physical activity in male mice (Xu et al, 2015). In our preparations, increased physical activity was closely associated with elevated estrogen levels in POMC[Cre:Cas9-GFP] female mice receiving retroAAV-*Magel2* sgRNA injection to the MeA. Moreover, we recently showed that most of the ER-α–positive cells in the MeA also expressed MC4Rs (Kwon & Jo, 2020). It is plausible that estrogen would regulate locomotor activity via activation of ER-α and MC4R–co-expressing neurons in the MeA. However, central infusion of the broad ER-α antagonist ICI 182,780 failed to block the effect of the loss of *Magel2* in ARC[POMC] neurons innervating the MeA on locomotor activity. Instead, ICI 182,780 infusion increased locomotor activity in female POMC[Cre:Cas9-GFP] mice receiving retroAAV-*Magel2* sgRNA injection to the MeA.

Although this ER-α antagonist has been widely used to treat breast cancer in postmenopausal women, this drug is known to activate GPER. Increased locomotor activity only in POMC[Cre:Cas9-GFP] female mice receiving retroAAV-*Magel2* sgRNA injection to the MeA might be due to activation of GPER in the brain. Indeed, treatment with the GPER antagonist effectively blocked the effect of the loss of the *Magel2* gene in POMC neurons on locomotor activity in female mice. GPER was found in the hypothalamus, including the VMH and the MeA in both male and female rodents (Hazell et al, 2009; Marraudino et al, 2021). As ER-α in these brain structures promoted physical activity in mice (Ogawa et al, 2003; Xu et al, 2011, 2015; Krause et al, 2021), ER-α and GPER may coordinately or independently regulate locomotor activity and eventually, energy balance. In fact, treatment with the GPER agonist caused a reduction of body weight in ovariectomized female mice and prevented body weight gain in male DIO mice (Sharma et al, 2020). Although plasma estrogen levels in male mice were outside of the detection range in our preparations, it is possible that GPER may play a role in controlling locomotor activity and energy balance in male mice as well. In fact, aromatase that converts testosterone into estrogen was highly expressed in the MeA and the number of aromatase-positive cells in the MeA was higher in males than in females (Wu et al, 2009). In addition, GPER was expressed in the MeA in both males and females (Llorente et al, 2020). Thus, a local increase in estrogen levels in the MeA may activate GPER in male mice as well. However, we could rule out the possibility that estrogen-independent mechanisms may contribute to the control of body weight and locomotor activity in our animal model. Collectively, deletion of the *Magel2* gene in the ARC[POMC]→MeA neural circuit elevated estrogen levels, which may result in activation of central GPER in female mice. This would promote physical activity, causing body weight

---

the *Magel2* gene in ARC[POMC] neurons innervating the MeA (POMC[Cre], n = 5 mice; POMC[Cre:Cas9-GFP], n = 6 mice). Two-tailed *t* test, *P < 0.05; **P < 0.01. **(K, L, M)** Summary plot showing TEE between the experimental groups. Regression plot showing TEE versus body weight (M).

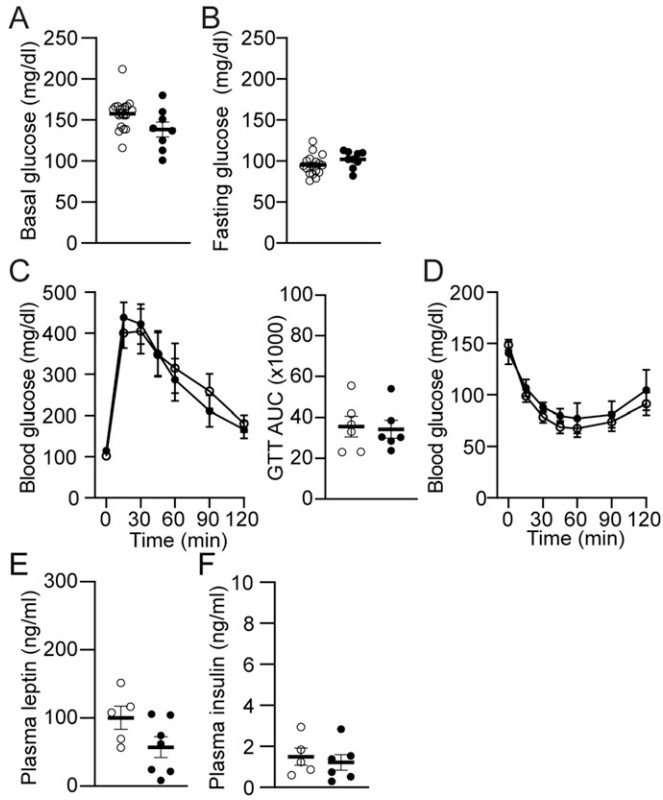

**Figure 5. Loss of the *Magel2* gene in ARC^POMC neurons innervating the MeA does not change glucose homeostasis in female mice fed with HFD.**
**(A, B)** Summary plots showing non-fasting and fasting blood glucose levels in POMC^Cre (open circle) and POMC^Cre:Cas9-GFP (closed circle) mice receiving retrograde AAV-*Magel2* sgRNA viral injection to the MeA (POMC^Cre, n = 19 mice and POMC^Cre:Cas9-GFP, n = 8 mice for non-fasting blood glucose; POMC^Cre, n = 17 mice and POMC^Cre:Cas9-GFP, n = 9 mice for fasting blood glucose). **(C)** Pooled data showing GTT in mice with and without the *Magel2* gene in ARC^POMC neurons innervating the MeA (left). Right panel: graphs showing AUC values obtained from GTT experiments (two-tailed *t* test, POMC^Cre, n = 6 mice; POMC^Cre:Cas9-GFP, n = 6 mice). **(D)** Pooled data showing ITT in mice with and without the *Magel2* gene in ARC^POMC neurons innervating the MeA. No significant difference was observed between the groups. **(E, F)** Summary plots showing levels of plasma leptin and insulin in mice with and without the *Magel2* gene in ARC^POMC neurons innervating the MeA (POMC^Cre, n = 5 mice, POMC^Cre:Cas9-GFP, n = 7 mice for leptin; POMC^Cre, n = 5 mice, POMC^Cre:Cas9-GFP, n = 6 mice for insulin).

loss in female mice. Hence, our studies may provide a novel therapeutic strategy for treating excessive body weight gain in individuals with PWS.

# Materials and Methods

## Ethics statement

All mouse care and experimental procedures were approved by the Institutional Animal Care Research Advisory Committee of the Albert Einstein College of Medicine and were performed in accordance with the guidelines described in the NIH guide for the care and use of laboratory animals. Stereotaxic surgery and viral injections were performed under isoflurane anesthesia.

## Animals

Mice used in this study included POMC-Cre (stock # 005965), floxed-stop Cas9-eGFP (stock # 026175), and floxed-stop Rosa26-eGFP mice (stock # 004077) that we purchased from the Jackson Laboratory. Both female and male mice of mixed C57BL/6J, FVB, and 129 strain backgrounds were used. Animals were housed in groups in cages under conditions of controlled temperature (22°C) with a 12:12 h light–dark cycle and fed a standard chow diet with ad libitum access to water. After stereotaxic surgery, mice were given a high-fat diet (20% calories by carbohydrate, 20% by protein, and 60% by fat, 5.21 kcal/g, D12492; Research Diet) for 10 wk.

## Stereotaxic surgery and viral injections

To knock down the *Magel2* gene in ARC^POMC neurons innervating the MeA, retrograde AAV-PGK-loxp-tdTomato-loxp-U6-mouse *Magel2* sgRNA viruses (titer, 1 × 10^13 pfu/ml) were generated at Applied Biological Materials Inc (ABM). The sgRNAs were designed to target to the consensus coding sequence (CCDS) 52264.1 region of mouse *Magel2* (NM_013779.2). The sequences of *Magel2* sgRNA were the following: (1) sgRNA1: cgcagctaagtacgaatctg, (2) sgRNA2: gtagggcggc-tatggactgc, and (3) sgRNA3: atggtccaggctccaccgct.

6- and 7-wk-old mice (males, ~20 g and females, ~18 g) were anesthetized deeply with 3% isoflurane and placed in a stereotaxic apparatus (David Kopf Instruments). A deep level of anesthesia was maintained throughout the surgical procedure. Under isoflurane anesthesia (2%), retrograde AAV-PGK-loxp-tdTomato-loxp-U6-mouse *Magel2* sgRNA viruses (200 nl/per site, titer, 1 × 10^13 pfu/ml) were bilaterally injected into the MeA of POMC^Cre and POMC^Cre:Cas9+GFP (AP, −1.58 mm; ML, ± 2 mm; DV, −5 mm). A 2.5 μl Hamilton syringe, having a 33-G needle was used to inject a volume of 50 nl viruses every 5 min. The Hamilton syringe tip was left in place for 10 min after delivering viruses to prevent backflow of viral solution up the needle track.

## Measurement of food intake, body weight, body composition, and blood glucose levels

Mice were individually housed for a week for acclimation, and daily food intake was measured accurately manually by an investigator (regular weighing of food, control of spillage, and calculation of disappearance of grams) over 5 d at 10 wk post viral injection. Body weight was measured weekly at 9 AM. Body composition for fat mass and fat-free mass was assessed by ECHO MRI at our animal physiology core.

Blood samples were collected from the mouse tail, and a small drop of blood was placed on the test strip of a glucose meter. Non-fasting basal glucose levels were measured at 9:00 AM. Fasting blood glucose levels were measured after an overnight fast once at 10 wk.

## Assessment of glucose tolerance and insulin tolerance

For GTT, experimental and control mice at 10 wk post viral inoculation were fasted for 18 h (5:00 PM–11:00 AM). A sterile glucose solution was i.p. administered at a concentration of 2 g/kg (glucose/body weight) at time 0. The blood glucose levels were

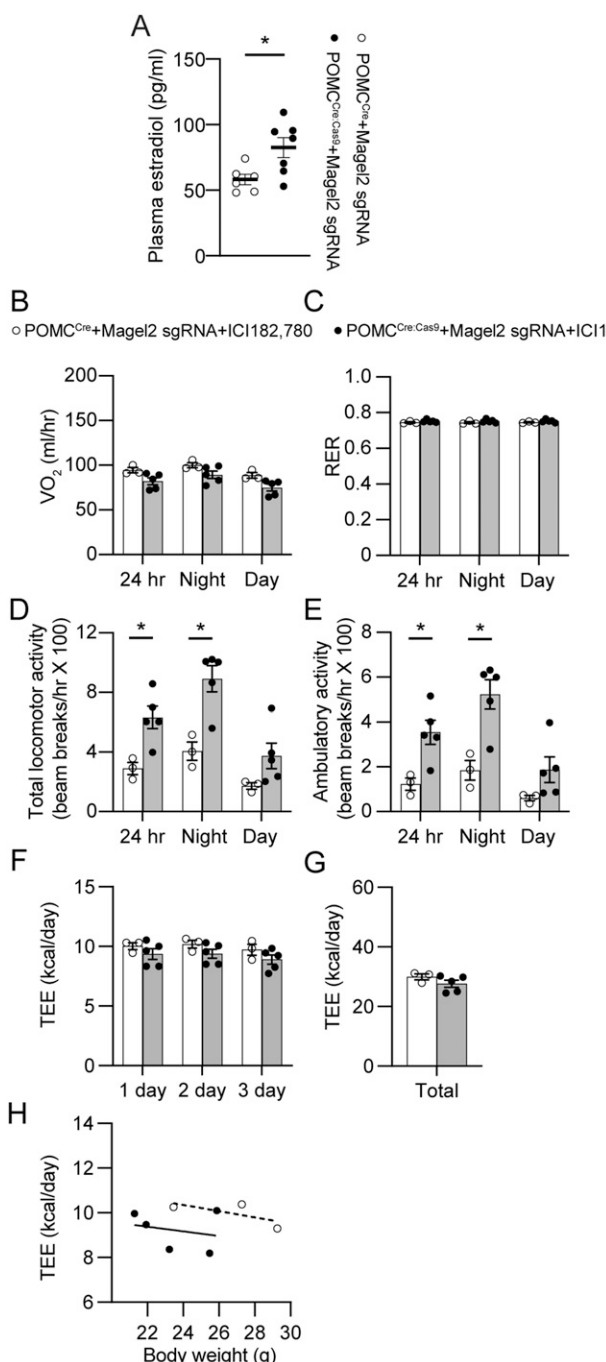

**Figure 6. Central ER-α does not contribute to the regulation of locomotor activity in female mice without the *Magel2* gene in ARC[POMC] neurons innervating the MeA.**

**(A)** Pooled data showing plasma estradiol levels between the groups. Estradiol levels were higher in female mice without the *Magel2* gene in ARC[POMC] neurons innervating the MeA than in controls (POMC[Cre], n = 6 mice; POMC[Cre:Cas9-GFP], n = 7 mice). Two-tailed test, \*$P < 0.05$. **(B, C)** Summary plots showing VO₂ and respiratory exchange ratio between the experimental groups with ICI 182,780 infusion. At 10 wk of viral injection, the ER-α antagonist ICI 182,780 (40 ng/day) was infused into the LV via osmotic pumps at a rate of 0.25 µl/hour. **(D, E)** Pooled data showing total and ambulatory activity between the experimental groups with ICI 182,780 infusion. An increase in locomotor activity was still observed in mice without the *Magel2* gene in ARC[POMC] neurons innervating the MeA after treatment with ICI 182,780 (two-tailed *t* test, night, \*$P < 0.05$; 24 h, \*$P < 0.05$). **(F, G, H)** Summary plot

measured at 15, 30, 60, 90, and 120 min after glucose injection. Blood glucose levels versus time after glucose injection were plotted, and the area under the curve was calculated and compared between the experimental and control groups.

For ITT, mice were fasted for 5 h (9:00 AM–2 PM). Blood glucose levels were measured at 0, 15, 30, 60, 90, and 120 min after i.p. injection of insulin (1 U/kg). We immediately injected glucose (2 g/kg) if the mice appeared ill because of insulin-induced hypoglycemia.

## ICV drug infusion

Mice were maintained under isoflurane anesthesia and placed in a stereotaxic apparatus. Under aseptic conditions, sterile guide cannulas were stereotaxically implanted into the lateral ventricle (AP, −0.22 mm; ML, +1 mm; DV −2.5 mm) for infusion of ICI 182,780 (40 ng/day, sc-203435A; Santa Cruz Biotechnology) or G15 (5 µg/day, 14673; Cayman). The guide cannulas were connected to osmotic pumps (1002W; RWD Life Science), and drugs were infused at a rate of 0.25 µl/hour for 10 d.

## Assessment of energy expenditure and locomotor activity

To examine if the loss of function of the *Magel2* gene in the ARC[POMC]→MeA circuit regulates energy expenditure and locomotor activity, we performed indirect calorimetry on mice fed with HFD for 10 wk. Mice were individually housed in the calorimeter cages and acclimated to the respiratory chambers for at least 2 d before gas exchange measurements. Indirect calorimetry was performed for 5 d at the end of 10 wk using an open-circuit calorimetry system. O₂ consumption and CO₂ production were measured for each mouse at 9-min intervals over a 24-h period. The RER was calculated as the ratio of CO₂ production over O₂ consumption. Locomotor activity in X-Y and Z planes was measured by infrared beam breaks in the calorimetry cages. All data were analyzed with a Web-based Analysis Tool for Indirect Calorimetry Experiments CalR (Mina et al, 2018) (version 1.3, https://calrapp.org/). An ANCOVA analysis was performed to determine if there was a significant difference in energy expenditure between the groups.

## Quantitative real-time PCR analysis

We collected ARC tissues from 9 AM to 10 AM without fasting. For qPCR analysis of the *Magel2* gene, total RNAs were isolated using the RNeasy mini kit (74104; QIAGEN) from ARC tissues, and then first-strand cDNAs were synthesized using the SuperScript III First-Strand synthesis kit (18080-051; Thermo Fisher Scientific). Real-time qPCR was performed in sealed 96-well plates with SYBR Green I master Mix (A25742; Applied Biosystems) using a Quant Studio 3 (Applied Biosystems). qPCR reactions were prepared in a final volume of 20 µl containing 2 µl cDNAs and 10 µl of SYBR Green master mix in the presence of primers at 0.5 µM. *β-actin* was used as an internal control for quantification of each sample. Amplification

showing TEE between the experimental groups. Regression plot showing TEE versus body weight (H).

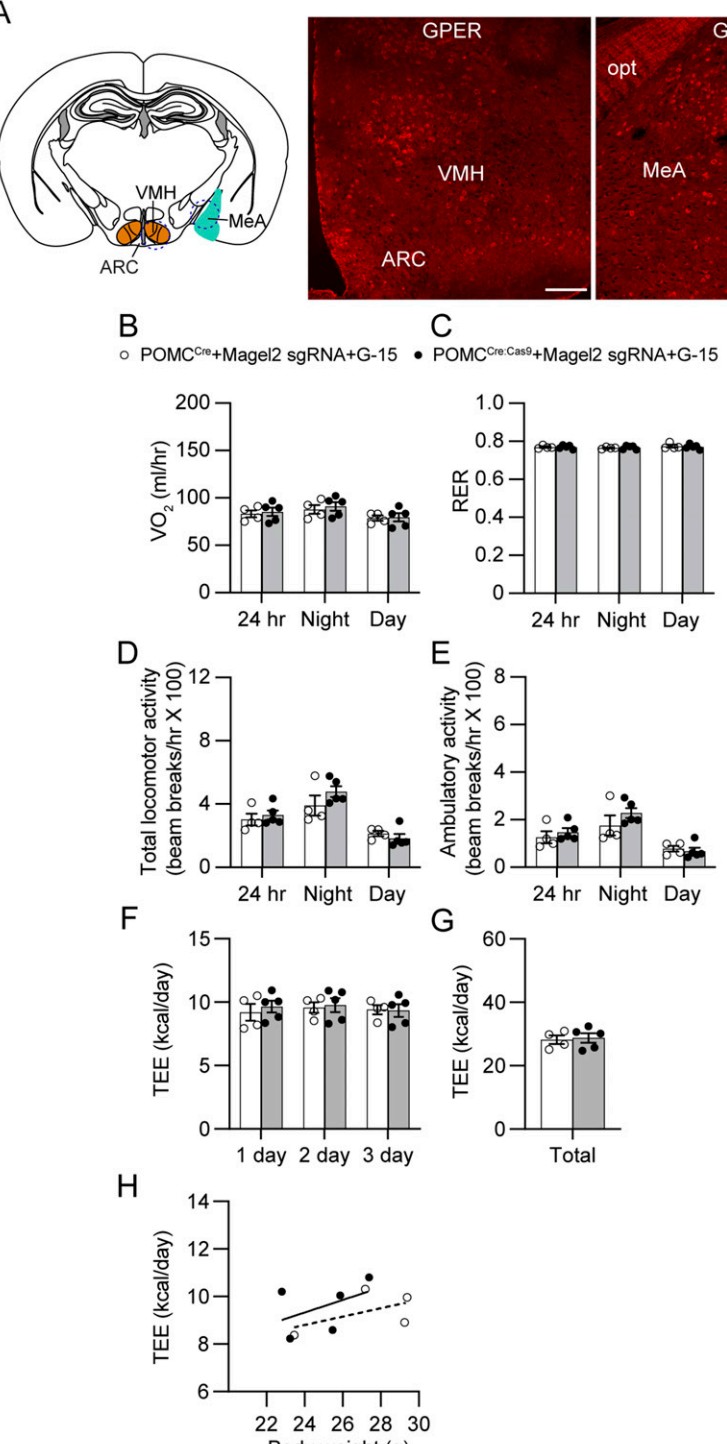

**Figure 7. Central GPER is critical in regulating locomotor activity in female mice without the *Magel2* gene in ARC^POMC neurons innervating the MeA.**
**(A)** Images of confocal fluorescence microscopy showing expression of GPER in the VMH (middle) and MeA (right) in female mice. Scale bar: 100 $\mu$m. **(B, C)** Summary plots showing $VO_2$ and respiratory exchange ratio between the experimental groups with G-15 infusion. At 10 wk of viral injection, the GPER antagonist G15 (5 $\mu$g/day [Mallet et al, 2021]) was infused into the LV via osmotic pumps at a rate of 0.25 $\mu$l/hour. There were no significant differences in $VO_2$ and respiratory exchange ratio between the experimental groups. **(D, E)** Pooled data showing total and ambulatory activity between the experimental groups with G-15 infusion. An increase in locomotor activity was not observed in mice without the *Magel2* gene in ARC^POMC neurons innervating the MeA following treatment with G-15 infusion. **(F, G, H)** Summary plot showing TEE between the experimental groups. Regression plot showing TEE versus body weight (H).

was performed under the following conditions: denaturation at 95°C for 30 s, followed by 40 cycles of denaturation at 95°C for 30 s, and annealing/extension at 60°C for 1 min. The primers used for qPCR were the following: *Magel2* forward, 5′-CAGCTCTCGGA-GATGGTAAATG-3′ and reverse, 5′-AAAGGTGCACTCCAGCTTAG-3′; and *β-actin* forward, 5′-CCTCTATGCCAACACAGTGC-3′ and reverse, 5′-GCTAGGAGCCAGAGCAGTAA-3′. The relative expression levels were determined using the comparative threshold cycle (CT), which was normalized against the CT of *β-actin* using the $^{\Delta\Delta}Ct$ method.

## Immunofluorescence staining

Mice were anesthetized with isoflurane (3%) and transcardially perfused with pre-perfusion solution (9 g NaCl, 5 g sodium nitrate,

10,000 U heparin in 1 liter distilled water) followed by 4% paraformaldehyde solution. Brains were removed and incubated in 4% paraformaldehyde overnight at 4°C and then placed into 30% sucrose solution for 2–3 d. Brain tissues were sectioned in 20 $\mu$m using a Lecia CM3050S cryostat. The sections were blocked in 0.1 M PBS buffer containing 0.2 M glycine, 0.1% triton X-100, 10% normal donkey serum, and 5% bovine serum albumin for 2 h at room temperature and then incubated with mouse anti-GFP (1:1,000, A-11120; Invitrogen), rabbit anti-MAGEL2 (1:2,000, a gift from Dr. Tacer [Chen et al, 2020]), rabbit anti-GPER (1:100, PA-528647; Invitrogen) antibodies for overnight at a cold room. And then, sections were washed three times in PBS and incubated with Alexa 488 anti-mouse IgG (1:500, Cat. no. A21202; Life Technologies), Alexa 568 anti-rabbit IgG (1:500, Cat. no. A10042; ; Life Technologies) for 2 h at room temperature. Tissues were washed, dried, and mounted with VECTASHIELD media containing DAPI. Images were acquired using a Leica SP8 confocal microscope. Cell counting was carried out with ImageJ software (version FIJI) as described in our previous work (Jeong et al, 2015, 2018; Kwon et al, 2020; Kwon & Jo, 2020). In brief, we used the Cell Counter plugin developed by Dr. Kurt De Vos (University of Sheffield). After initialization of the images, we manually countered MAGEL2+, GFP+, and MAGEL2/GFP+ cells in the ARC (250 × 250 $\mu$m).

### Measurement of plasma estradiol, testosterone, leptin, and insulin

Blood samples were collected from the retro-orbital plexus with heparinized capillary tubes (VWR International, LLC) and then centrifuged at 15,600$g$ for 10 min at 4°C to collect plasma. Plasma estradiol, testosterone, leptin, and insulin levels were quantified using the ELISA kits (Cayman Chemical, 501890 for estradiol; Cayman Chemical, 582701 for testosterone; Thermo Fisher Scientific, KMC2281 for leptin, and Mercodia, 10-1247-01 for insulin, respectively) according to the protocol provided by the manufacturer. Optical density was measured using a microreader.

### Statistics

All statistical results are presented as mean ± SEM. Statistical analyses were performed using GraphPad Prism 9.0. Two-tailed $t$ tests were used to calculate $P$-values of pair-wise comparisons. Time course comparisons between groups were analyzed using a two-way repeated-measures (RM) ANOVA with Sidak's correction for multiple comparisons. Data were considered significantly different when the probability value was less than 0.05.

## Supplementary Information

## Acknowledgements

We thank Drs. Patrick Potts and Klementina Fon Tacer for providing us with the MAGEL2 antibody and Drs. Gary Schwartz and Streamson Chua Jr. for their

valuable feedback and comments on this study. We also thank Dr. Shun-Mei Liu and Licheng Wu for their technical assistance. This work was supported by the NIH (RO1 DK092246, R01 AT011653, R03 TR003313, and P30 DK020541) and Foundation for Prader–Willi Research to Y-H Jo.

## Author Contributions

Y Choi: data curation, software, formal analysis, investigation, and methodology.
H-Y Min: data curation, investigation, and methodology.
J Hwang: data curation, formal analysis, investigation, and methodology.
Y-H Jo: conceptualization, data curation, formal analysis, supervision, funding acquisition, validation, investigation, methodology, project administration, and writing—original draft, review, and editing.

## Conflict of Interest Statement

The authors declare that they have no conflict of interest.

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
