## [Reviewer comments · Life Science Alliance]

Life Science Alliance

Magel2 knockdown in hypothalamic POMC neurons innervating the MEA reduces susceptibility to DIO.

Young-hwan Jo, Yuna Choi, Hyeon-Young Min, and Jiyeon Hwang

DOI: <https://doi.org/10.26508/lsa.202201502>

Corresponding author(s): Young-hwan Jo, Albert Einstein College of Medicine

Review Timeline:

Submission Date:	2022-04-25
Editorial Decision:	2022-06-03
Revision Received:	2022-07-25
Editorial Decision:	2022-08-12
Revision Received:	2022-08-12
Accepted:	2022-08-15

Transaction Report:

June 3, 2022

Re: Life Science Alliance manuscript #LSA-2022-01502

Dr. Young-hwan Jo
Albert Einstein College of Medicine
1300 Morris Park Ave
Forch 505
Bronx, NY 10461

Dear Dr. Jo,

Thank you for submitting your manuscript entitled "Magel2 knockdown in hypothalamic POMC neurons innervating the MEA prevents diet-induced obesity." to Life Science Alliance. The manuscript was assessed by expert reviewers, whose comments are appended to this letter. We invite you to submit a revised manuscript addressing the Reviewer comments.

Thank you for this interesting contribution to Life Science Alliance. We are looking forward to receiving your revised manuscript.

Sincerely,

B. MANUSCRIPT ORGANIZATION AND FORMATTING:

Reviewer #1 (Comments to the Authors (Required)):

The manuscript by Hyeon-Young Min demonstrated that CRISPR-deletion of *Magel2* from MeA-projecting POMC neurons in the ARC reduces susceptibility to diet-induced obesity. They also provide evidence that the beneficial metabolic effects of *Magel2* knockdown are attenuated after chronic brain ICV infusion of GPER1 antagonist G15. While the studies presented some interesting evidence to support the metabolic role of *Magel2* expressed by MeA-projecting POMC neurons in the ARC, some critical issues need to be further addressed to better support this model:

Majors:

1. Cas9/sgRNA targeting *Magel2* was used in the studies to delete *Magel2*. However, no details were provided in the Methods (sgRNA sequences, targeting genomic regions, possible off-target effects).
2. To confirm the neuroanatomical specificity of the POMC^{ARC}→MeA *Magel2* knockout approach, coimmunofluorescence staining would ensure that *Magel2* expression in the ARC is excluded from neurons that co-express the red (tdTOMATO from AAVrg) and green (Cas9-GFP) fluorescent reports (i.e., those POMC neurons with projections to the MeA which pick up the AAVrg-FLEX-tdTOMATO-sgRNA-*Magel2* vector)?
3. Although the authors propose an estrogen-mediated mechanism in females, no data or discussion was provided to explain the potential mechanism in males. Are there any changes in de novo synthesis of 17 β -estradiol in the brain (i.e., BNST, MeA, and POAH, which have high expression of aromatase in both sexes. PMID: 19804754)?
4. The description of chronic Osmatic Pump ICV infusion in the Methods was lacking details (concentration, vehicle, whether controls were infused with vehicle).
5. It is puzzling that mutant mice have decreased body weight, associated with normal food intake and energy expenditure. The body weight balance is eventually determined by the homeostasis between energy intake and energy expenditure. Physical activity is part of the expenditure, which cannot explain the body weight phenotype when energy expenditure is normal. The energy expenditure may be subtle and needs to analyze further using ANCOVA (PMID: 22205519).
6. No detailed description of ImageJ software (version Fiji) cell counting was found in the cited papers (Jeong et al 2015, Jeong et al 2018, Kwon & Jo 2020, Kwon et al 2020). The representative images in Figure 1A (2nd upper panel, MAGL2 in male POMC-Cre:*Rosa26*-GFP) and 1B (2nd upper panel, MAGL2 in female POMC-Cre:*Rosa26*-GFP) do not match with the representative images in Figure 2B (Left image showing MAGL2 in POMC-Cre male) and Figure 4B (Left image showing MAGL2 in POMC-Cre female). The actual numbers of POMC-Cre:*Rosa26*-GFP, MAGEL2 positive neurons, and dual-color neurons should be provided in Figure 1C to facilitate the comparison between Figure 1C and 2B (right dot plot) or 4B (right dot plot).
7. The actual weight of fat and lean mass should be included in Figures 2 and 4.
8. In Figures 6 and S2, it would make more sense to compare POMCCre+*Magel2* sgRNA and POMCCre:Cas9+*Magel2* sgRNA first. Then to test whether metabolic effects induced by *Magel2* knockdown can be blocked by ICI 182, 780 (POMCCre+*Magel2* sgRNA+ICI182,780 vs. POMCCre:Cas9+*Magel2* sgRNA+ICI182,780), or G15 (POMCCre+*Magel2* sgRNA+G-15 and POMCCre:Cas9+*Magel2* sgRNA+G-15).

Minors:

1. Page 4, para 2. The authors pointed out that "There was a significant difference in the MAGEL2 expression in ARCPOMC neurons between male and female POMCCre:*Rosa26*-GFP miceand" then stated that "These results support the interpretation XXX". I couldn't follow a direct logic to make this statement.
2. Page 6, Para 3, Line 2. Authors mentioned "Both groups consumed a similar amount of chow". Apparently, it should be "Both groups consumed a similar amount of HFD".

Reviewer #2 (Comments to the Authors (Required)):

This manuscript by Dr. Young-Hwan Jo and colleagues reports novel and exciting observations that knocking down *Magel2* in ARC-POMC neurons that innervate the medial amygdala (MeA) increases dark phase locomotor activity and reduces body weight gain in HFD-fed mice. This knockdown was achieved using a retrograde AAV-*Magel2* sgRNA injected into the MeA of POMC-Cre:Cas9-eGFP mice, or in POMC-Cre mice as controls. The authors have experience using CRISPR-Cas9 technology and previously validated other gene knockdown in ARC-POMC neurons. In female mice with *Magel2* knockdown in ARC-POMC projecting to the MeA, circulating estrogen levels are increased, and central pharmacological blockade of GPER (but not of central ER-alpha) lowers locomotor activity, although it is not clear if this affects body weight gain, or changes in fat and/or lean mass and if male mutant mice similarly utilize central GPER despite no detectable levels of circulating estrogen.

Centrally mediated regulation of mechanisms that control systemic energy homeostasis is a relevant topic and important given the growing diabetes and obesity epidemic. The manuscript demonstrates a coherent progression of ideas. A few revisions may help clarify some concerns, particularly with references to *Magel2* deletion/knockdown as "PWS models". Some comments are listed below.

Major comments

1. It is interesting that blocking nuclear ER-alpha, which was expected to block *Magel2* knockdown effects, increased rather than lowered physical activity (Suppl Fig 2). Is this a different cohort than in Fig 4 and 5? Is there confirmation of *Magel2* KD, and/or data showing that this dose of ER-alpha inhibitor can indeed block E2 action?
2. The GPER findings are exciting. Did central G15 administration, which lowered locomotor activity, affect body weight in experiments shown in Fig 6?
3. Is it possible to genetically block or knockdown GPER vs ER-alpha? If not experimentally, perhaps the authors can comment? This could help to determine the specificity of the estrogen receptor effects. Since the ER-alpha inhibitor can also activate GPER, it is possible that blocking ER-alpha, without simultaneous GPER activation, may also have an effect in female mutant mice.
4. A few of the sentences in the second paragraph of the discussion are a little confusing and seem to indicate that *Magel2* deficiency both leads to increased body weight and also protection from obesity. Is this diet-/age-/sex- specific? What about increased fat mass and high blood insulin/leptin in the following sentences? Please clarify.

Minor comments

5. At what timepoint were ARC tissues collected for qPCR analyses? Fasting or fed, and/or after an experiment?
6. On page 5 and 6 of the PDF manuscript regarding the statements, "...were protected from DIO" and "protected against DIO" - is there non-DIO to compare with? If not, perhaps "lowered body weight gain" is more appropriate.
7. In the final sentence of the results section (PDF page 8), the authors may wish to revise the statement ("Taken together, our results suggest that central GPER rather than ER-alpha is required for promoting locomotor activity in PWS mouse models") since the *Magel2* knockdown occurs here in just a subpopulation of cells, and these animals were not hyperphagic. Also, the GPER data currently describes female mice.
8. Twice in the first 4 sentences of the discussion section - typographical - should it instead read "... ARC-POMC innervating MeA..." (and not vice versa)?
9. The summary statement near the end of the first paragraph of the discussion section: "Finally, blockade of GPER but not ER-alpha completely abolished the effect of loss of the *Magel2* gene in ARC-POMC neurons innervating the MeA." Perhaps it would be more accurate to indicate that this was observed in female mice only.
10. Given the reported similarity of the male mutant mice to *Snord116*-deficient mice (e.g. reduced body, no effect on food intake, decreased %fat mass, greater activity at night), can the authors comment on, or provide data to show, whether a lack of *Magel2* in ARC POMC neurons innervating MeA also reduces *Snord116*?
11. The sentence in the final paragraph of the Discussion starting with, "Collectively, deletion of the *Magel2* gene in..." should specify "in female mice". An additional set of experiments in male mutant mice, which did not show elevated estrogen levels, in which central GPER are activated, would be required as a start to suggest the role of GPER in a more generalized way.

Reviewer #3 (Comments to the Authors (Required)):

Min and colleagues submitted an interesting manuscript to the journal investigating the role of *magel2* knockdown in hypothalamic POMC neurons innervating the medial amygdala. They demonstrate that in both male and female mice, removal of *magel2* using *crispr-cas9* mediated ablated using retrograde AAVs in the medial amygdala reduces the susceptibility to diet-induced obesity likely via reduced locomotor activity. The paper would be of interest to the journal, but several issues should be addressed prior to publication.

-The authors used a POMC-Cre transgene strategy to study hypothalamic POMC cells. However, Lori Zeltser published a paper (not referenced) where her lab demonstrated that *cre*-activity of POMC is outside of the arcuate nucleus using this model. Without an Arcuate specific stereotaxic injection, *cre* activity is in many places in the brain and periphery, as referenced in this paper. For one, this brings into question whether the transgene containing cells really are POMC-containing. Antibody for a peptide within the cell (ACTH or α -MSH) should have been used with colchicine treated animals rather than this approach due to the issues laid out in this paper, among others with specificity of *cre* activity. Secondly, *Cre* activity will be in places outside of the arcuate. For instance, they note that *cre* activity has been observed in the amygdala, which bring into question the specificity of arcuate POMC cells throughout the paper because sgRNA was targeted for the amygdala and will not just target arcuate POMC cells. A more careful inspection and discussion is needed at the absolute minimum, but more investigation is needed to support the conclusions raised by the authors that these phenomena are only from hypothalamic connections to the amygdala, when *cas9* activity will be many other places in the brain.

-Moreover, the authors injected 200nl per side and details of the injections are minimal. Due to the issues in the previous concern, more is needed to describe what was done. What did the spread of the injections look like? Was there a *gfp* reporter? Where did the *gfp* cells extend to? Did the authors measure *magel2* levels in all areas that contain POMC-*cre* activity? How fast were the injections made? Did the authors check for damage to the injection site? Was the AAV toxic? The authors should create a supplemental figure to demonstrate the spread of the injections throughout the cases and examples of what these injections looked like. In addition, the authors should provide additional details, age, weight, speed, stereotax used, injection method, and AAV titers.

-In the male mice, the groups look different prior to AAV injection. It is not clear why the authors did not have a *Cas9*-POMC-*cre* control. Was the *Cas9* on a different background from POMC-*cre*, leading to these mice being on a slightly lower developmental trajectory? It brings into question whether the mice are starting off at a different place because they didn't control for this potential issue that can play a role in the susceptibility to diet induced obesity.

-The metabolic chamber data was expressed as a function of body weight, but body weight is different. Normalization factor should not be different between groups. Even though lean % is different, is total lean mass different between groups? If not, the authors could use ANCOVA to analyze this data. The data presented in this way brings the question that the authors may be missing an elevation in energy expenditure in the mice, potentially independent of locomotor activity.

-If analyzed properly and there is no difference in energy expenditure, the authors should discuss this finding. This would be surprising since increasing body weight typically leads to a decrease in energy expenditure, if measured as a function of body weight but not when either no normalization or lean body mass are used to normalize the data.

-The conclusions in the last figure are not supported by the data. Since the animals already have a difference in activity, this is not the appropriate way to measure the response to estrogen. The authors should have included vehicle treated mice to measure the effect of estrogen and used that to compare the groups. However, measuring responses following the GPER agonist may just lead to a different response independent of estrogen action. The authors should be careful to make this interpretation because they base the whole discussion on these findings, which are not well supported. At the least, the authors should broaden their discussion off factors other than estrogen action, which may not be the only mediator at play.

We thank the Editor and expert Reviewers for their careful and detailed review of the manuscript. In revising the manuscript, we have addressed all the concerns raised by reviewers. Our responses and revisions to the manuscript are detailed in the rebuttal letter.

Responses to Reviewer #1:

Majors:

RE: 1. Cas9/sgRNA targeting *Mage12* was used in the studies to delete *Mage12*. However, no details were provided in the Methods (sgRNA sequences, targeting genomic regions, possible off-target effects).

-. We now included this information in the Methods and Materials section (page 13).

RE: 2. To confirm the neuroanatomical specificity of the POMC^{ARC}→MeA *Mage12* knockout approach, coimmunofluorescence staining would ensure that *Mage12* expression in the ARC is excluded from neurons that co-express the red (tdTOMATO from AAVrg) and green (Cas9-GFP) fluorescent reports (i.e., those POMC neurons with projections to the MeA which pick up the AAVrg-FLEX-tdTOMATO-sgRNA-*Mage12* vector)?

-. We completely agree with the Reviewer. For that purpose, we designed and generated our retrograde AAV-PGK-FLEX-tdTomato-U6-mouse *Mage12* sgRNA. Under these experimental conditions, we expect that the U6 promoter induces *Mage12* sgRNA expression, and the PGK promoter drives Cre-mediated tdTomato expression in infected ARC^{POMC} neurons. To be honest, despite our efforts, we failed to get what we expected. In fact, prior studies (PMID, 27288456 & 23341784) described that a loss of the *Mage12* gene induced cell death, particularly ARC^{POMC} neurons. Likewise, we found that the number of ARC^{POMC} neurons in POMC^{Cre-Cas9} receiving retroAAV-*Mage12* sgRNA injection into the MeA was much fewer than that in controls at 10 weeks post viral injections (Ref. Fig 1A and B). For this reason, tdTomato/GFP-positive cell bodies were very scarce in the ARC of POMC^{Cre-Cas9} receiving retroAAV-*Mage12* sgRNA injection into the MeA. Instead, we could observe tdTomato-positive fibers exclusively in the MeA and ARC (Ref. Fig. 1C and D), indicating that our viral injection and sgRNA expression were appropriate. Occasionally, tdTomato/POMC-positive cells were detected in the median eminence (Ref. Fig. 1E). Hence, instead of performing double immunostaining, we decided to quantify the *Mage12* knockdown using RT-qPCR and immunohistochemistry with an anti-MAGEL2 antibody. At the end of the experiments, the expression of the *Mage12* gene in the ARC was systematically confirmed by RT-qPCR analysis. When the viral injections were missed, we excluded the data.

Reference Figure 1. Loss of the *Mage12* gene in ARC^{POMC} neurons projecting to the MeA reduces the number of POMC neurons in the ARC. **A** and **B.** Images showing α -MSH-positive cells in the ARC of POMC^{Cre-Cas9} mice receiving AAV-scrambled sgRNA (**A**) or AAV-*Mage12* sgRNA (**B**) injection to the MeA 10 weeks after viral injections. The number of α -MSH-positive cells in the ARC was fewer in POMC^{Cre-Cas9} mice receiving *Mage12* sgRNA injection than in controls. **C** and **D.** Images showing tdTomato-expressing structures such as the MeA and ARC in the brain of POMC^{Cre-Cas9} mice receiving *Mage12* sgRNA injection in the MeA. **E.** Images showing ARC^{POMC} neurons infected with retroAAV-FLEX-tdTomato-U6-*Mage12* sgRNA.

RE: 3. Although the authors propose an estrogen-mediated mechanism in females, no data or Discussion was provided to explain the potential mechanism in males. Are there any changes in *de novo* synthesis of 17 β -estradiol in the brain (i.e., BNST, MeA, and POAH, which have high aromatase expression in both sexes. PMID: 19804754)?

-. Thanks for the suggestions, Accordingly, we revised the Discussion (page 12).

Yes, there may be changes in *de novo* synthesis of 17 β -estradiol, particularly in the MeA. The Shah research group elegantly showed that aromatase converting testosterone into estrogen was highly expressed in the MeA, and the number of aromatase-positive cells in the MeA was higher in males than in females (PMID: 19804754). In addition, GPER was expressed in the MeA in both males and females (PMID: 32849310). Thus, a local increase in estrogen levels in the MeA may activate GPER in male mice.

RE: 4. The description of chronic Osmotic Pump ICV infusion in the Methods was lacking details (concentration, vehicle, whether controls were infused with vehicle).

-. We added this information in the Materials and Methods section (page 14). We also re-analyzed our results, as suggested by the Reviewer. We first compared POMC^{Cre+Magel2 sgRNA} with POMC^{Cre:Cas9+Magel2 sgRNA} (Fig. 2I and J; Fig. 4I and J). And then, we examined whether metabolic effects induced by *Magel2* knockdown were blocked by ICI 182, 780 (new Fig. 6C and D), POMC^{Cre+Magel2 sgRNA+ICI182,780} vs. POMC^{Cre:Cas9+Magel2 sgRNA+ICI182,780}, or G15 (New Fig. 7E and F, POMC^{Cre+Magel2 sgRNA+G-15} vs. POMC^{Cre:Cas9+Magel2 sgRNA+G-15}).

RE: 5. It is puzzling that mutant mice have decreased body weight, associated with normal food intake and energy expenditure. The body weight balance is eventually determined by the homeostasis between energy intake and energy expenditure. Physical activity is part of the expenditure, which cannot explain the body weight phenotype when energy expenditure is normal. The energy expenditure may be subtle and needs to analyze further using ANCOVA (PMID: 22205519).

-. As suggested by the Reviewer, we re-analyzed our indirect calorimetry results using a Web-based Analysis Tool for Indirect Calorimetry Experiments CalR (version 1.3, <https://calrapp.org/>, Mina et al., 2018, Cell Metab). Although we failed to detect a “significant” difference in energy expenditure between the two groups, we don’t think that there was no difference in energy expenditure between the groups during high-fat feeding. It is highly possible that our system may not be able to detect a subtle change in energy expenditure. It is also possible that an accumulation of small improvements in basal energy expenditure due to increased physical activity from the beginning of high-fat feeding may prevent weight gain in our animal model. In fact, female mice lacking the *Magel2* in ARC^{POMC} neurons innervating the MeA exhibited a trend toward an increase in total energy expenditure (Fig. 4K).

In our current study, we measured energy expenditure for 5 days at the end of 10 weeks on high-fat feeding. Our future study will measure changes in energy expenditure over a period of time (at 1, 3, 5, 8, and 10 weeks post-viral injections while fed HFD.), which will provide a more solid explanation of how the *Magel2* knockdown in ARC^{POMC} neurons innervating to the MeA regulates body weight.

RE: 6. No detailed description of ImageJ software (version Fiji) cell counting was found in the cited papers (Jeong et al 2015, Jeong et al 2018, Kwon & Jo 2020, Kwon et al 2020). The representative images in Figure 1A (2nd upper panel, MAGL2 in male POMC-Cre:Rosa26-GFP) and 1B (2nd upper panel, MAGL2 in female POMC-Cre:Rosa26-GFP) do not match with the representative images in Figure 2B (Left image showing MAGL2 in POMC-Cre male) and Figure 4B (Left image showing MAGL2 in POMC-Cre female). The actual numbers of POMC-Cre:Rosa26-GFP, MAGEL2 positive neurons, and dual-color neurons should be provided in Figure 1C to facilitate the comparison between Figure 1C and 2B (right dot plot) or 4B (right dot plot).

-. We now included this information and added new figures (Fig. 1B and D). We also added a detailed description of cell counting in the Methods and Materials section (pages 15-16)

RE: 7. The actual weight of fat and lean mass should be included in Figures 2 and 4.

-. As the reviewer suggested, we changed Figures 2D & E and 4D & E and revised the text accordingly.

RE: 8. In Figures 6 and S2, it would make more sense to compare POMCCre+Magel2 sgRNA and POMCCre:Cas9+Magel2 sgRNA first. Then to test whether metabolic effects induced by *Magel2* knockdown can be blocked by ICI 182, 780 (POMCCre+Magel2 sgRNA+ICI182,780 vs. POMCCre:Cas9+Magel2 sgRNA+ICI182,780), or G15 (POMCCre+Magel2 sgRNA+G-15 and POMCCre:Cas9+Magel2 sgRNA+G-15).

-. We re-analyzed the results, as suggested by the Reviewer (new Fig. 6 and 7). We appreciate the Reviewer’s comments that help us improve the quality of our manuscript.

Minors:

RE: 1. Page 4, para 2. The authors pointed out that "There was a significant difference in the MAGEL2 expression in ARCPOMC neurons between male and female POMCCre:Rosa26-GFP mice and" then stated that "These results support the interpretation XXX". I couldn't follow a direct logic to make this statement.

-. We are sorry for the confusion. We revised the text (page 9).

RE: 2. Page 6, Para 3, Line 2. Authors mentioned "Both groups consumed a similar amount of chow". Apparently, it should be "Both groups consumed a similar amount of HFD".

-. We now changed it to HFD.

Responses to Reviewer #2:

Major comments

RE: 1. It is interesting that blocking nuclear ER-alpha, which was expected to block Magel2 knockdown effects, increased rather than lowered physical activity (Suppl Fig 2). Is this a different cohort than in Fig 4 and 5? Is there confirmation of Magel2 KD, and/or data showing that this dose of ER-alpha inhibitor can indeed block E2 action?

-. We chose the dose based on previous studies, in which that dose of the ER- α antagonist completely blocked the effect of central ER- α (PMID:17142339; PMID: 18599599; PMID:19123998). Hence, we expect that the treatment should block central ER- α .

No, that was the same cohort shown in Fig. 4 and 5.

RE: 2. The GPER findings are exciting. Did central G15 administration, which lowered locomotor activity, affect body weight in experiments shown in Fig 6?

-. Yes, we are also very excited about our findings. In our current study, the GPER antagonist appeared to increase body weight (Ref. Fig. 2) after 10 days of infusion, associated with reduced physical activity. We will carefully examine this important issue in our near future study.

Reference figure 2. Effect of the GPER antagonist on body weight.

RE: 3. Is it possible to genetically block or knockdown GPER vs. ER-alpha? If not experimentally, perhaps the authors can comment? This could help to determine the specificity of the estrogen receptor effects. Since the ER-alpha inhibitor can also activate GPER, it is possible that blocking ER-alpha, without simultaneous GPER activation, may also have an effect in female mutant mice.

-. This is an excellent suggestion. Technically, it is possible to knock down ER- α in the MeA using the ER- α siRNA. The Xu research group demonstrated that males lacking ER- α in the MeA significantly gained more weight than the control group when given HDF and exhibited reduced physical activity during the dark phase (PMID: 26098212). Hence, we think that ER- α and GPER in the MeA may coordinately or independently regulate locomotor activity and, eventually, energy balance. We will also investigate this important issue in our future study.

RE: 4. A few of the sentences in the second paragraph of the Discussion are a little confusing and seem to indicate that Magel2 deficiency both leads to increased body weight and also protection from obesity. Is this diet-/age-/sex- specific? What about increased fat mass and high blood insulin/leptin in the following sentences? Please clarify.

-. We are sorry for the confusion. We revised the text (page 9).

Minor comments

RE: 5. At what timepoint were ARC tissues collected for qPCR analyses? Fasting or fed, and/or after an experiment?

-. We collected the tissues from 9 am to 10 am without fasting.

RE: 6. On pages 5 and 6 of the PDF manuscript regarding the statements, "...were protected from DIO" and "protected against DIO," - is there non-DIO to compare with? If not, perhaps "lowered body weight gain" is more appropriate.

-. As suggested, we revised them (page 5 and 6).

RE: 7. In the final sentence of the results section (PDF page 8), the authors may wish to revise the statement ("Taken together, our results suggest that central GPER rather than ER-alpha is required for promoting locomotor activity in PWS mouse models") since the Magel2 knockdown occurs here in just a subpopulation of cells, and these animals were not hyperphagic. Also, the GPER data currently describes female mice.

-. As suggested, we changed the text (page 8)

RE: 8. Twice in the first 4 sentences of the discussion section - typographical - should it instead read "... ARC-POMC innervating MeA..." (and not vice versa)?

-. We are sorry about it. We now changed our typographical errors.

RE: 9. The summary statement near the end of the first paragraph of the discussion section: "Finally, blockade of GPER

but not ER-alpha completely abolished the effect of loss of the *Magel2* gene in ARC-POMC neurons innervating the MeA." Perhaps it would be more accurate to indicate that this was observed in female mice only.

-. We now stated that this was observed in female mice only.

RE: 10. Given the reported similarity of the male mutant mice to *Snord116*-deficient mice (e.g., reduced body, no effect on food intake, decreased %fat mass, greater activity at night), can the authors comment on, or provide data to show, whether a lack of *Magel2* in ARC POMC neurons innervating MeA also reduces *Snord116*?

-. The resistance of DIO in mice lacking *Magel2* exclusively in ARC^{POMC} neurons innervating the MeA is somewhat unexpected and in contrast with the phenotypes observed in the PWS population. As the Reviewer noted, our findings are consistent with prior studies showing that mice lacking the *Magel2* and *Snord116* genes are DIO-resistant. Hence, the effects of loss of function of individual genes in the PWS domain may differ from those of a loss of the entire imprinted genes in the PWS domain. Each gene in the PWS imprinted region may have a distinct function and differently contribute to the phenotypes of PWS. In addition, individual genes expressed in different brain areas also appear to have different functions. In fact, adult deletion of *Snord116* in the ventromedial hypothalamus promoted hyperphagia, and a subset of these mutant mice developed obesity. Hence, it is necessary to define the role of each gene in the PWS domain, which will significantly improve our understanding of the pathophysiology of PWS. Given that the role of individual genes in the PWS domain remains understudied, we will investigate if a lack of *Magel2* in ARC^{POMC} neurons innervating the MeA also reduces *Snord116* in ARC^{POMC} neurons and other brain areas in our near future study.

RE: 11. The sentence in the final paragraph of the Discussion starting with, "Collectively, deletion of the *Magel2* gene in..." should specify "in female mice". An additional set of experiments in male mutant mice, which did not show elevated estrogen levels, in which central GPER are activated, would be required as a start to suggest the role of GPER in a more generalized way.

-. We revised the final paragraph of the Discussion based on the comments from you and Reviewer #1.

Responses to Reviewer #3:

RE: 1. The authors used a POMC-Cre transgene strategy to study hypothalamic POMC cells. However, Lori Zeltser published a paper (not referenced) where her lab demonstrated that cre-activity of POMC is outside of the arcuate nucleus using this model. Without an Arcuate specific stereotaxic injection, cre activity is in many places in the brain and periphery, as referenced in this paper. For one, this brings into question whether the transgene containing cells really are POMC-containing. Antibody for a peptide within the cell (ACTH or α -MSH) should have been used with colchicine treated animals rather than this approach due to the issues laid out in this paper, among others with specificity of cre activity. Secondly, Cre activity will be in places outside of the arcuate. For instance, they note that cre activity has been observed in the amygdala, which bring into question the specificity of arcuate POMC cells throughout the paper because sgRNA was targeted for the amygdala and will not just target arcuate POMC cells. A more careful inspection and Discussion is needed at the absolute minimum, but more investigation is needed to support the conclusions raised by the authors that these phenomena are only from hypothalamic connections to the amygdala, when cas9 activity will be many other places in the brain.

- Yes, we agree entirely with the Reviewer. We are also aware of the work of the Zeltser research group. They demonstrated that POMC-expressing immature cells could switch off the *Pomc* gene and gave rise to NPY/AgRP neurons during gestation and that the POMC^{Cre} transgenic mice exhibited the expression of the Cre recombinase outside of the ARC, including the MeA. However, its expression levels in the MeA were scarce.

To address this important issue, we performed three independent experiments in our prior (PMID: 33250721) and ongoing studies. First, we examined if ARC^{POMC} neurons “directly” send axonal projections to the MeA using a Cre-dependent anterograde AAV having the GFP reporter protein (Ref. Fig.3. A-C) and the POMC^{Cre} transgenic mice. We found that a subset of ARC^{POMC} neurons projected to the MeA and that GFP-positive fibers in the MeA were indeed positive to ACTH. Second, we injected a Cre-dependent retrograde AAV having the tdTomato reporter protein into the MeA of POMC^{Cre} mice. As shown in Ref. Fig. 3D-G, all the tdTomato-positive cells in the ARC expressed GFP, indicating that they are POMC neurons. Importantly, we did not observe tdTomato-positive cells outside of the ARC. Third, we injected a retrograde AAV encoding the Cre recombinase under the control of neuronal POMC enhancers into the MeA of floxed-stop Cas9-eGFP mice. We found that all the GFP-expressing neurons in the ARC expressed POMC (Ref. Fig. 3H-I). Based on these findings, we think that almost all the neurons infected with AAV in our preparations are ARC^{POMC} neurons innervating the MeA.

Reference Figure 3. Feasibility, Efficacy, reproducibility, and scientific premise of our experimental approaches. **A.** Schematic illustration showing the experimental configurations (left panel). A Cre recombinase-inducible anterograde viral tracer AAV1-FLEX-GFPsm was injected into the ARC of the POMC^{Cre} mice. Right panel: The ARC of the POMC^{Cre} mice injected with AAV1-FLEX-GFPsm exhibited GFP-positive cells in the ARC. Scale bar: 30 μ m **B.** Image of fluorescence confocal microscopy showing GFP-positive fibers and terminals in the MeA of the POMC^{Cre} mice injected with AAV1-FLEX-GFPsm. opt, optic tract, Scale bar: 100 μ m **C.** Images of fluorescence confocal microscopy showing co-expression of GFP and ACTH in axonal fibers in the MeA (white arrow). Scale bar: 20 μ m **D.** Schematic illustration showing the experimental configurations (top panel). POMC^{Cre;eGFP} mice were stereotaxically injected with a Cre-inducible retrograde viral tracer retroAAV-FLEX-tdTomato into the MeA. TdTomato-positive fibers were observed in the MeA (bottom panel). Scale bar: 200 μ m **E.** Images of fluorescence confocal microscopy showing retrogradely identified ARC^{POMC} neurons projecting to the MeA. Retrogradely identified cells (white arrow) were found in the ipsilateral, but not contralateral, ARC of the POMC^{Cre;eGFP} mice injected with retroAAV-FLEX-tdTomato. Importantly, most tdTomato-positive cells were found in the lateral part of the ARC. Scale bar: 50 μ m **F.** Images of fluorescence confocal microscopy showing retrogradely labeled ARC^{POMC} neurons. Most tdTomato neurons co-expressed GFP in the POMC^{Cre;eGFP} mice injected with retroAAV-FLEX-tdTomato into the MeA (white arrow). Scale bar: 30 μ m **G.** Schematic drawing showing the location of retrogradely identified cells (x) in the ARC. 3V, 3rd ventricle **H** and **I.** Images of confocal fluorescence microscopy showing Cre-mediated GFP expression in ARC^{POMC} neurons projecting the MeA. Arrowheads represent MeA-projecting ARC^{POMC} neurons.

While ARC^{POMC} neurons have been strongly implicated in controlling energy balance and glucose homeostasis, they displayed substantial heterogeneity in their neurochemical signals, membrane receptors, neurophysiological responses, projection sites, and gene expression profiles. Recent single-cell RNA sequencing studies described more than 30 different POMC neuron subtypes. In addition to neuropeptides, ARC^{POMC} neurons produce and release fast neurotransmitters, including GABA and glutamate, and a small population that co-releases both GABA and glutamate. Due to this heterogeneity of ARC^{POMC} neurons, acute stimulation of the whole population of ARC^{POMC} neurons did not change food intake (PMID, 23426689; PMID, 27869800; PMID, 21209617), despite the well-known anorectic effects of one of their neuropeptides, α -MSH. For this reason, we did **NOT** inject viruses into the ARC. In other words, we did NOT knock down the *Magel2* gene in the entire population of ARC^{POMC} neurons. Instead, we used **projection-specific approaches** to knock down the *Magel2* gene exclusively in neuroanatomically distinct ARC^{POMC} neurons. In fact, our prior study showed that the ARC^{POMC}->MeA pathway regulated acute food intake (PMID, 33250721).

It is also important to note that the retrograde AAV we used is *replication-incompetent*. This replication-incompetent retrograde AAV can't cross synapses. AAV particles are taken up by axon terminals, and then AAV is transported retrogradely to cell bodies. Hence, we did not see GFP expression outside of the ARC (Ref. Fig. 2D-G; PMID, 33250721).

RE: 2 Moreover, the authors injected 200nl per side, and details of the injections are minimal. Due to the issues in the previous concern, more is needed to describe what was done. What did the spread of the injections look like? Was there a gfp reporter? Where did the gfp cells extend to? Did the authors measure magel2 levels in all areas that contain POMC-cre activity? How fast were the injections made? Did the authors check for damage to the injection site? Was the AAV toxic? The authors should create a supplemental figure to demonstrate the spread of the injections throughout the cases and examples of what these injections looked like. In addition, the authors should provide additional details, age, weight, speed, stereotax used, injection method, and AAV titers.

- We are sorry for the lack of details. We now described all the details that the Reviewer requested in the Methods and Materials section.

As the AAV lacks apparent pathogenicity, AAV has been used for gene transfer in animal models and clinical trials. In other words, no safety concerns for AAV vectors have been reported. Importantly, our immunostaining results showed Cre expression in the ARC but unfortunately failed to observe Cre expression in the amygdala, including the medial and basomedial structures. This can be due to different experimental conditions, such as age, sex, homozygous vs. heterozygous Cre transgene, number of animals used, etc. In fact, the study of the Zeltser research group did not provide any such information (PMID, 22166984). In fact, the Tonegawa research group showed that β -galactosidase (a Cre-loxP recombination marker) expression was restricted to the hippocampus and ARC of 12-week-old POMC^{Cre;Rosa26} transgenic mice (Ref. Fig. 4: McHugh et al., 2007, Science). Note that both groups used the same POMC^{Cre} transgenic strain.

Reference Figure 4. Cre-induced β -Gal expression in the brain of POMC^{Cre} mice. Image from McHugh et al., 2007, Science

Additionally, as the MeA did not receive synaptic inputs from the dentate gyrus (PMID, 25503449), where the Cre expression was observed in POMC^{Cre} mice, our viral vectors could not express in the dentate gyrus.

We designed and generated our retrograde AAV-PGK-FLEX-tdtomato-U6-mouse *Magel2* sgRNA. This viral vector has the FLEX Cre-On switch to create an **inducible, turn-on** tdTomato system under the PGK promoter. Hence, the expression of tdTomato was restricted to Cre-positive cells and axon terminals. Indeed, the reporter protein tdTomato was expressed exclusively in the MeA, to which ARC^{POMC} neurons projected and the ARC (Ref. Fig. 5B and C). More importantly, the *Magel2* gene is primarily expressed in the hypothalamus and MeA in mice (Ref. Fig. 6, PMID, 32879135). In other words, *Magel2* knockdown will occur exclusively in those areas.

Reference Figure 5. A. Schematic illustration of our experimental configurations. **B and C.** Images showing the tdTomato expression in the MeA and the ARC.

Reference Figure 6. *Magel2* expression in the hypothalamus and MeA (Image from Chen et al., 2020, JCI insight)

RE: 3 In the male mice, the groups look different prior to AAV injection. It is not clear why the authors did not have a Cas9-POMC-cre control. Was the Cas9 on a different background from POMC-cre, leading to these mice being on a slightly lower developmental trajectory? It brings into question whether the mice are starting off at a different place because they didn't control for this potential issue that can play a role in the susceptibility to diet induced obesity.

- Both are on a C57BL/6 genetic background. Hence, we don't think that the genetic background influences our results. We re-analyzed body weight data in male mice by removing three outliers from 20 mice (>3 times SEM). We added a new figure (Fig. 2C).

We used the POMC^{Cre} strain as a control because we wanted to ensure that our viral vector did not affect ingestive behavior. As shown in Ref. Fig. 7, we also injected AAV having scrambled sgRNA into the MeA of male POMC^{Cre}-Cas9 mice (n= 8 mice) and found no difference in body weight between the two control groups during high-fat feeding.

Reference Figure 7. Body weight gain in POMC^{Cre}+AAV-*Magel2* sgRNA (n= 17 mice, open circle) and POMC^{Cre}-Cas9+AAV-scrambled sgRNA (n=8 mice, filled circle). There was no significant difference in body weight between the two groups during high-fat feeding.

RE: 4. The metabolic chamber data was expressed as a function of body weight, but body weight is different. Normalization factor should not be different between groups. Even though lean % is different, is total lean mass different between groups? If not, the authors could use ANCOVA to analyze this data. The data presented in this way brings the question that the authors may be missing an elevation in energy expenditure in the mice, potentially independent of locomotor activity.

-. As suggested, we re-analyzed our indirect calorimetry results using a Web-based Analysis Tool for Indirect Calorimetry Experiments CalR (version 1.3, <https://calrapp.org/>, Mina et al., 2018, Cell Metab) (Fig. 2 and 4). Although we failed to detect a “significant” difference in energy expenditure, we think that this does not mean that there was no difference in energy expenditure between the groups during high-fat feeding. In our study, we measured energy expenditure for 5 days at the end of 10 weeks on high-fat feeding. It is highly possible that our system may not be able to detect a subtle change in energy expenditure. It is also possible that an accumulation of tiny improvements in basal energy expenditure from the beginning of high-fat feeding may prevent weight gain in our animal model. In fact, female mice lacking the *Magel2* in ARC^{POMC} neurons innervating the MeA exhibited a trend toward an increase in total energy expenditure (Fig. 4K). Our future study will measure changes in energy expenditure over a period of time (at 1, 3, 5, 8, and 10 weeks post-viral infections while fed HFD.), which will provide a more solid explanation of how the *Magel2* knockdown in ARC^{POMC} neurons innervating to the MeA regulates body weight.

RE: 5. If analyzed properly and there is no difference in energy expenditure, the authors should discuss this finding. This would be surprising since increasing body weight typically leads to a decrease in energy expenditure, if measured as a function of body weight but not when either no normalization or lean body mass are used to normalize the data.

-. See above (RE:4)

RE: 6. The conclusions in the last figure are not supported by the data. Since the animals already have a difference in activity, this is not the appropriate way to measure the response to estrogen. The authors should have included vehicle treated mice to measure the effect of estrogen and used that to compare the groups. However, measuring responses following the GPER agonist may just lead to a different response independent of estrogen action. The authors should be careful to make this interpretation because they base the whole Discussion on these findings, which are not well supported. At the least, the authors should broaden their Discussion off factors other than estrogen action, which may not be the only mediator at play.

-. We completely agree with the Reviewer. Reviewer #1 also had a similar comment. Hence, we now re-analyzed our results and included new figures 6 and 7. In addition, we also revised our discussion (pages 11-12).

August 12, 2022

RE: Life Science Alliance Manuscript #LSA-2022-01502R

Dr. Young-Hwan Jo
Albert Einstein College of Medicine
1300 Morris Park Ave
forch 505
Bronx, NY 10461

Dear Dr. Jo,

Thank you for submitting your revised manuscript entitled "Magel2 knockdown in hypothalamic POMC neurons innervating the MEA prevents diet-induced obesity.". We would be happy to publish your paper in Life Science Alliance pending final revisions necessary to meet our formatting guidelines.

- please address Reviewer 1 and 2's remaining comments
- please make sure the author order in your manuscript and our system match
- Figure 2B needs a scale bar

A. FINAL FILES:

B. MANUSCRIPT ORGANIZATION AND FORMATTING:

****It is Life Science Alliance policy that if requested, original data images must be made available to the editors. Failure to provide**

original images upon request will result in unavoidable delays in publication. Please ensure that you have access to all original data images prior to final submission.**

The license to publish form must be signed before your manuscript can be sent to production. A link to the electronic license to publish form will be sent to the corresponding author only. Please take a moment to check your funder requirements.

Sincerely,

Reviewer #1 (Comments to the Authors (Required)):

Most of the concerns have been well addressed.

It is recommended to include the figures/justifications provided for questions 2 and 5 in the results/discussion.

"2. To confirm the neuroanatomical specificity of the POMC^{ARC}→MeA Magel2 knockout approach, coimmunofluorescence staining would ensure that Magel2 expression in the ARC is excluded from neurons that coexpress the red (tdTOMATO from AAVrg) and green (Cas9-GFP) fluorescent reports (i.e., those POMC neurons with projections to the MeA which pick up the AAVrg-FLEX-tdTOMATO-sgRNA-Magel2 vector)?"

"5. It is puzzling that mutant mice have decreased body weight, associated with normal food intake and energy expenditure. The body weight balance is eventually determined by the homeostasis between energy intake and energy expenditure. Physical activity is part of the expenditure, which cannot explain the body weight phenotype when energy expenditure is normal. The energy expenditure may be subtle and needs to analyze further using ANCOVA (PMID: 22205519)."

Reviewer #2 (Comments to the Authors (Required)):

Thank you for the clarification and revisions. My minor request is for some of the responses provided in the responses to reviewers, which pertain to methods, be incorporated in the text. While minor revisions, these details are felt to be important. These include:

- (1) Re: 1 In lieu of data showing that the dose of ER-alpha inhibitor used blocks central ER-alpha, please include rationale and citations for using the 40ng/day dose icv (based on previous studies cited in the responses to reviewers); and
- (2) Re:5 - non-fasting status of mice when tissues were collected for analysis.

Otherwise, the questions and concerns from my initial review have been addressed.

Reviewer #3 (Comments to the Authors (Required)):

The authors addressed my concerns.

We thank the Editor and expert Reviewers for their careful and detailed review of the manuscript. In revising the manuscript, we have addressed all the concerns raised by reviewers. Our responses and revisions to the manuscript are detailed in the rebuttal letter.

Responses to Reviewer #1:

Most of the concerns have been well addressed.

It is recommended to include the figures/justifications provided for questions 2 and 5 in the results/discussion.

Re: 2. To confirm the neuroanatomical specificity of the POMCARC→MeA Magel2 knockout approach, coimmunofluorescence staining would ensure that Magel2 expression in the ARC is excluded from neurons that coexpress the red (tdTOMATO from AAVrg) and green (Cas9-GFP) fluorescent reports (i.e., those POMC neurons with projections to the MeA which pick up the AAVrg-FLEX-tdTOMATO-sgRNA-Magel2 vector)?"

-. We now included Fig. S1 as the reviewer suggested.

Re: 5. It is puzzling that mutant mice have decreased body weight, associated with normal food intake and energy expenditure. The body weight balance is eventually determined by the homeostasis between energy intake and energy expenditure. Physical activity is part of the expenditure, which cannot explain the body weight phenotype when energy expenditure is normal. The energy expenditure may be subtle and needs to analyze further using ANCOVA (PMID: 22205519)."

-. We revised the discussion section (page 11).

Responses to Reviewer #2:

Thank you for the clarification and revisions. My minor request is for some of the responses provided in the responses to reviewers, which pertain to methods, be incorporated in the text. While minor revisions, these details are felt to be important. These include:

Re: 1 In lieu of data showing that the dose of ER-alpha inhibitor used blocks central ER-alpha, please include rationale and citations for using the 40ng/day dose icv (based on previous studies cited in the responses to reviewers).

-. We now added this information the main text (page 7)

Re:5 - non-fasting status of mice when tissues were collected for analysis.

-. We also included this information in the materials and methods section (page 15)

August 15, 2022

RE: Life Science Alliance Manuscript #LSA-2022-01502RR

Dr. Young-Hwan Jo
Albert Einstein College of Medicine
1300 Morris Park Ave
forch 505
Bronx, NY 10461

Dear Dr. Jo,

Thank you for submitting your Research Article entitled "Magel2 knockdown in hypothalamic POMC neurons innervating the MEA reduces susceptibility to DIO.". It is a pleasure to let you know that your manuscript is now accepted for publication in Life Science Alliance. Congratulations on this interesting work.

DISTRIBUTION OF MATERIALS:

Again, congratulations on a very nice paper. I hope you found the review process to be constructive and are pleased with how the manuscript was handled editorially. We look forward to future exciting submissions from your lab.

Sincerely,
